# SMPLY PRIVATE: FROM MASKS TO MESHES IN ACTION RECOGNITION

## ABSTRACT

In this paper, we introduce MASK2MESH (M2M), a novel privacy-preserving data augmentation framework that effectively bridges the realism gap seen in synthetic-based action recognition methods. Traditional privacy-enhancing techniques, such as feature masking and synthetic data supplementation, tend to degrade data quality and reduce model performance. In contrast, our method leverages the SMPL-X model to replace real humans with detailed 3D meshes in video data, preserving the subtle nuances of human movement and expressions that are crucial for accurate action recognition. By augmenting real data with superimposed meshes, M2M simplifies both pre-training and fine-tuning processes, without introducing the overheads and biases typically associated with synthetic data. Empirical results show that our approach achieves performance within 0.5% of models trained on unmodified video data, proving that overlaying meshes leads to no significant performance loss in action recognition tasks. This work presents a practical solution for data anonymization without compromising accuracy, offering valuable insights for more efficient and scalable video data processing techniques in computer vision and action recognition.

## 1 INTRODUCTION

Action recognition, the process of classifying human activities based on video sequences, is crucial for applications such as surveillance, human-computer interaction, and video analytics (Herath et al., 2017). Traditional action recognition systems rely heavily on extensively annotated datasets to achieve optimal performance. With advancements in deep learning and the emergence of vision transformers (ViTs) (Dosovitskiy et al., 2021), pre-training models on large datasets has become standard practice to enhance accuracy and generalization (Pham et al., 2022). However, these datasets often include identifiable individuals, raising significant privacy and ethical concerns (Yoo & Choi, 2013).

Data sharing, particularly without obtaining explicit consent from individuals, necessitates robust de-identification methods. Conventional anonymization techniques, such as blurring and pixelation, often degrade data quality, thereby reducing its efficacy for action recognition tasks. Moreover, these methods rely on heuristics and may not effectively balance privacy protection with data utility (Ren et al., 2018). Ensuring individual privacy while sharing video data can significantly advance research and development in action recognition and computer vision, where large datasets are imperative. Privacy-preserving techniques that maintain data quality can enhance the accuracy and reliability of machine learning models, facilitating more robust and fair applications. Nonetheless, these methods do not fully address the realism gap introduced by synthetic methods, nor do they completely safeguard privacy, as variable visuals like skin tone and gender can still be discerned (Zhong et al., 2023; Dave et al., 2022; Li et al., 2022a).

Our research aims to address these challenges by demonstrating that using mesh bodies can effectively remove biases and close the realism gap in action video recognition while preserving privacy. In particular, we: **(1)** investigate whether meshes can preserve privacy while approximating real-world data and reducing the realism gap; **(2)** explore the potential of using meshes to mitigate biases related to background, scene-object interactions, race, and gender, examining whether they can serve as a standardization technique for the human form.

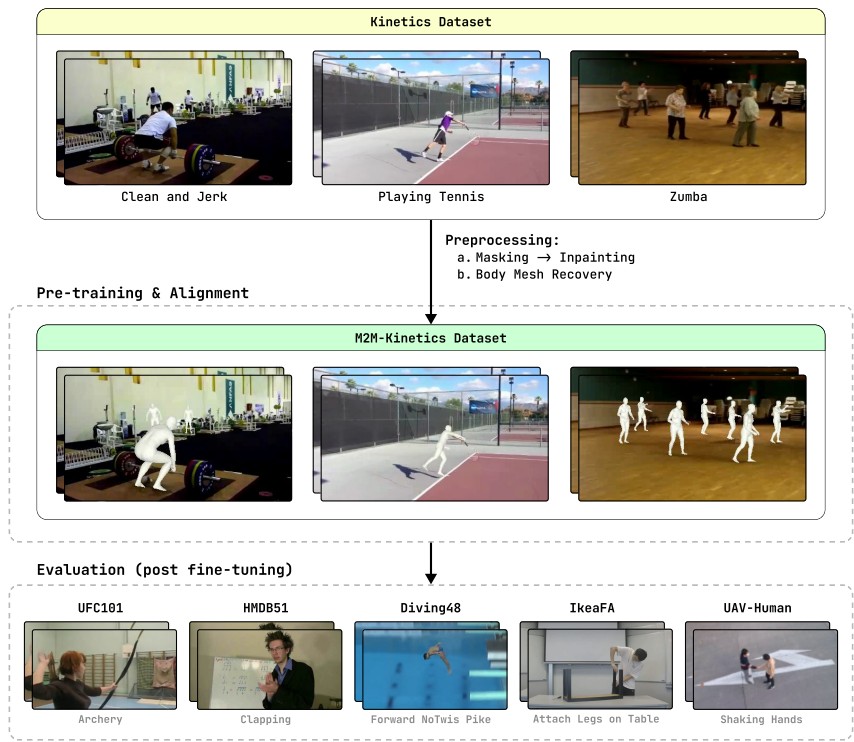

Figure 1: **SMPLy Private in Action.** Transforming human actions into privacy-preserving 3D meshes: videos from the Kinetics dataset are preprocessed using masking, inpainting, and body mesh recovery to replace humans with 3D mesh bodies. The M2M-augmented videos are then used for pre-training and alignment, with their performance evaluated across various human action recognition tasks, demonstrating the framework's ability to maintain high fidelity and ensure ethical data usage.

To do so, we introduced a unified framework, *SMPLy Private*, that combines mesh-based pretraining and supervised label-alignment stages to address privacy concerns and maximize action recognition performance. Unlike prior methods, our approach integrates mesh representations into both pretraining and alignment, ensuring that privacy-preserving augmented features are fully leveraged across training phases. Specifically:

* *Pre-training*: We use mesh-based augmented data for masked autoencoder (MAE) (He et al., 2021) pre-training, focusing on reconstructing masked video features while incorporating human motion dynamics.

* *Alignment*: The same mesh representations are used during the supervised alignment phase to ensure consistency and maximize transferability to downstream tasks.

This dual-stage process, ensures that the benefits of privacy-preserving data extend seamlessly across training phases, addressing the limitations of prior approaches that treat pre-training and alignment as a slightly more disjoint processes.

Our approach is motivated by the premise that mesh bodies can replace humans in video data, maintaining privacy. However, to illustrate our interest in exploring the impact of mesh bodies on mitigating gender and race biases, consider an example. Video footage from the National Basketball Association (NBA) in the 1980s-1990s, predominantly features African-American players. Training models on these videos to predict contemporary players, who are more diverse, could introduce bias by predominantly associating basketball with African-American individuals. This concern extends to gender bias as well (Burns et al., 2019). Employing meshes could standardize player representation across different demographics, thereby reducing biases and promoting ethical model development.

Our contributions are summarized as follows:

**(1)** We introduce MASK2MESH (M2M), a framework that replaces real humans in video datasets with detailed 3D meshes using the SMPL-X (Pavlakos et al., 2019) model, effectively preserving privacy without compromising the integrity of action recognition. Our method outperforms existing privacy-preserving benchmarks and rivals non-privacy-preserving methods, closing the realism gap.

**(2)** By strategically modifying the k-means clustering algorithm (Lloyd, 1982), we introduce K-NEXUS, a dataset sampling strategy designed to eliminate action class bias. This enhancement significantly improves the performance of M2M.

**(3)** Our targeted study on gender representation in 3D meshes reveals that gender-neutral meshes improve average performance in action recognition tasks within gender-biased classes, suggesting that neutral representations can effectively mitigate gender biases while maintaining privacy and ensuring fairness in automated video analysis.

**(4)** We demonstrate that models fine-tuned on M2M-augmented data learn representations quicker due to consistent mesh depictions, offering a new perspective on bridging the realism gap and serving as a standardization technique for ethical model development across diverse demographic groups.

## 2 RELATED WORKS

**Preservation of Privacy in Action Recognition.** Anonymization techniques, such as blurring, down-sampling, adversarial augmentations, masking faces and other identifiable features, are commonly adapted strategies to maintain human confidentiality (Dai et al., 2015; Butler et al., 2015; Piergiovanni & Ryoo, 2020; Zhang et al., 2021; Dave et al., 2022; Wu et al., 2020; Pittaluga et al., 2019) but can be left susceptible to revealing an individual's identity based on characteristics such as color and size (Yang et al., 2022; Oh et al., 2016). We eliminate the human form and replace it with 3D meshes, thus curating training data that does not have explicitly identifiable forms or features of the original human.

**Body Mesh Recovery.** Estimating 3D human body poses and shapes is a complex challenge addressed by various methods. SMPL-X uses a unified 3D model trained on extensive 3D scans, providing detailed and realistic representations. HybrIK-X (Li et al., 2023) effectively combines 3D keypoint estimation with body mesh reconstruction by converting precise 3D joint locations into relative body-part rotations. However, its application is limited to single individuals, which is not suitable for videos featuring multiple people. In contrast, our approach utilizes OSX (Lin et al., 2023), which excels in multi-human mesh recovery. OSX employs a unified encoder-decoder architecture integrated with a component-aware transformer. This setup not only predicts body parameters but also enhances segmentation, crucial for accurate face and hand estimation. By eliminating the need for separate networks and manual post-processing, OSX provides more natural and plausible 3D meshes. Given its simplicity and effectiveness in handling complex scenes with multiple individuals, OSX is our chosen method for accurate human body mesh recovery for M2M.

**Biases and Synthetic Data.** Object-scene bias in video action recognition refers to the tendency of models to rely on static objects or backgrounds rather than the dynamic actions themselves for classification (Yun et al., 2020). To address this issue, various augmentation strategies such as loss augmentation (Choi et al., 2019), action-scene swapping (Zou et al., 2022), and video compositing (Gowda et al., 2022) are prevalent but not privacy-preserving. One method mitigates this bias by first learning background information from real data and then temporal information from entirely synthetic data rather than augmenting components of the actual video (Zhong et al., 2023). Although this approach is privacy-preserving and focuses on learning background and actions, thereby addressing object-scene bias, the use of synthetic videos still leaves a desire for the realism gap to be bridged (Friedman et al., 2023). Instead, our approach returns to augmenting real videos by masking, in-painting, and overlaying appropriate mesh bodies in place of the original human. The plain "mannequin-like" mesh bodies remove any discriminatory bias, unlike the synthetic "video-game-like" humans, which tend to have features such as hair color, gender, skin tone, etc.

**Self-Supervised Pretraining in Action Recognition.** The training scheme for action recognition models is crucial to the performance on downstream tasks, given that most data in nature is unlabeled. Self-supervised learning (SSL) has proven to be a powerful pretraining mechanism in such schemes (Balestriero et al., 2023). Furthermore, the default choice of encoder has shifted from convolutional neural networks (CNNs) like temporal segment networks (TSNs) (Wang et al., 2016) and inflated

3D convolutional networks (I3Ds) (Carreira & Zisserman, 2018) to vision transformers (ViTs) (Dosovitskiy et al., 2021), as they effectively process frames as patch sequences, enabling the capture of long-range dependencies and patterns in videos. SSL is typically categorized into four paradigms: deep metric learning (Chen et al., 2020; Dwibedi et al., 2021; Koohpayegani et al., 2020; Khosla et al., 2020), self-distillation (Grill et al., 2020; Chen & He, 2021; Caron et al., 2021), canonical correlation analysis (Bardes et al., 2021; Zbontar et al., 2021; Caron et al., 2020; Ermolov et al., 2021), and masked image modeling (He et al., 2021; Xie et al., 2022; Chang et al., 2023). We chose the latter paradigm, employing a masked autoencoder (MAE) training scheme, specifically using VideoMAE (Wang et al., 2022), which typically incorporates the base vision transformer architecture (ViT-B) and has previously achieved state-of-the-art performance on benchmarks like UCF101 (Soomro et al., 2012) and Kinetics (Kay et al., 2017).

## 3 METHODS

### 3.1 TRANSITIONING FROM SYNTHETIC TO AUGMENTED

SMPLy Private marks a significant transition over current state-of-the-art approaches like SynAPT and PPMA, which rely predominantly on fully synthetic data for privacy preservation (Kim et al., 2022; Zhong et al., 2023). PPMA also follows a two-stage process: MAE pre-training on human-removed data, then supervised alignment using synthetic and human-removed datasets. However, its reliance on video game-like synthetic datasets (i.e, SURREACT, PHAV, ElderSim (Varol et al., 2021; De Souza et al., 2017; Hwang et al., 2023)), which modify entire scenes and objects alongside subjects, limits their ability to capture nuanced, real-world contextual features present in action recognition datasets like Kinetics (Jordon et al., 2024; Hao et al., 2024), thereby hindering downstream transferability. M2M overcomes this limitation by employing mesh-based augmentation to synthesize privacy-preserving action data while retaining real-world scenes and objects. Instead of creating entirely new environments, M2M overlays parametric human motion meshes over removed human subjects, preserving contextual and environmental nuances while obfuscating identifiable human features. This approach bridges the realism gap, yielding richer temporal and contextual representations without compromising privacy. The core distinction lies in data treatment. While PPMA integrates synthetic and human-obfuscated real data to enhance temporal and contextual understanding, its dependency on synthetic datasets introduces domain shifts away from real-world distributions. In contrast, M2M maintains real-world contexts by embedding privacy-preserving meshes, achieving balanced representation. Furthermore, M2M extends alignment by integrating mesh representations directly into pretraining, seamlessly incorporating temporal and contextual features for improved downstream transferability.

### 3.2 DATASET CURATION: K-NEXUS

We use a subset of the Kinetics-400 (Kay et al., 2017) video dataset as our training dataset where we select 150 classes amongst the 400 along with at most 1,000 videos per class. Previous works (Kim et al., 2022; Zhong et al., 2023) have used manual or random splits to curate their custom dataset (Kinetics-150), however, we consider the action class bias in the Kinetics dataset (e.g., actions like playing violin and playing guitar are visually closer than playing volleyball). To obtain discrete classes and reduce the bias, we uniquely deploy a k-means clustering algorithm (Lloyd, 1982) to obtain the final set of classes. Our approach aims to assemble a $K$-class dataset $D^*$ of minimal bias from an existing dataset $D$ with $C$ classes. The class labels of $D$ are denoted by $L = \{l_1, l_2, \ldots, l_C\}$, where $l_i$ represents the $i$-th class of $D$. Our objective is to find a subset $L^* = \{l_1^*, \ldots, l_K^*\}$ such that: $l_i^* \in L$, $l_i^* \neq l_j^*$ for $i \neq j$, and $D^*$ has minimal bias. Specifically, we perform the following steps for label sampling:

**(1) Encoding action image-label pairs.** Let $V$ be the set of all videos and $L$, previously defined, be the set of all class labels. $V_{l_j} \subseteq V$ is then the set of all videos with class label $l_j \in L$. For each video $v_i \in V_{l_j}$, we sample a random frame $I_{v_i}$ from $v_i$'s mid-quartile range. We then construct a set of tuples $\Theta = \{\theta_{v_i, l_j} \mid v_i \in V, l_j \in L\}$ where $\theta_{v_i, l_j} = (I_{v_i}, l_j)$. An embedding function is then defined as $f : \theta \to \mathbb{R}^d$, which maps the tuple $\theta$ to a $d$-dimensional embedding space using a LLaVA image-text encoder (Liu et al., 2023) to accommodate both the image and its corresponding text label. The embedding of a tuple $\theta_{v_i, l_j}$ is given by: $\mathbf{e}_{v_i, l_j} = f(\theta_{v_i, l_j})$. To compute

the average embedding for class $l_j$, we aggregate the embeddings for the frame-label pairs from all videos in $V_{l_j}$: $E_{l_j} = \{\mathbf{e}_{v_i, l_j} \mid v_i \in V_{l_j}\}$. Then, the average embedding $\bar{\mathbf{e}}_{l_j}$ for class $l_j$ is given by: $\bar{\mathbf{e}}_{l_j} = \frac{1}{|V_{l_j}|} \sum_{\mathbf{e} \in E_{l_j}} \mathbf{e}$. The set of average embeddings for all class labels is then given by: $\bar{E} = \{\bar{\mathbf{e}}_{l_j} \mid l_j \in L\}$.

**(2) K-means clustering.** We then apply a modified k-means clustering algorithm (see Appendix B.1), that minimizes dataset bias instead of the within-cluster sum of squares, to partition the embeddings in $\bar{E}$ into $K$ clusters. Let $\kappa(\bar{E}, K)$ denote the modified clustering operation, resulting in cluster assignments $\Omega = \{\omega_1, \omega_2, \ldots, \omega_C\}$, where each $\omega_i \in \{1, 2, \ldots, K\}$.

**(3) Selecting representative labels.** For each cluster $k \in \{1, 2, \ldots, K\}$, we identify the labels that belong to cluster $k$. Let $L_k = \{l_i \mid \omega_i = k\}$ be the set of labels in cluster $k$. From each cluster $k$, we select the representative label $l_k^*$ with the highest number of video samples within the cluster: $l_k^* = \arg\max_{l_i \in L_k} |V_{l_i}|$ where $V_{l_i}$ is the set of all videos with class label $l_i$ and $|V_{l_i}|$ denotes the number of videos in $V_{l_i}$. The final set of labels $L^*$ is then given by: $L^* = \{l_k^* \mid k \in \{1, 2, \ldots, K\}\}$. Thereby obtaining $D^*$ (Kinetics-150). Our method, termed k-Means Neural Embeddings eXploited for Unified Sampling (K-NEXUS), groups similar actions together and selects representative labels to reduce action class bias in Kinetics[1]. By doing so, K-NEXUS ensures a more balanced set of labels and minimizes representation entropy bias across different action categories.

## 3.3 MASK2MESH AUGMENTATION

Our proposed M2M augmentation framework is designed to achieve privacy-preserving video data by replacing real humans in videos with 3D mesh representations while preserving essential motion details. The framework consists of two main modules: (1) the masking and inpainting module; (2) the body mesh recovery module. We leverage the Kinetics-150 dataset curated in the aforementioned Section 3.2, resizing all video clips to $432 \times 240$ to standardize the input data. The first step in our framework involves detecting and removing human figures from the video frames. This process is divided into two sub-tasks: human detection and in-painting. We utilize the Segment Anything Model (SAM) (Kirillov et al., 2023), to generate masks for human figures in each video frame. The generated masks are passed on to the subsequent inpainting module, which involves filling the regions occupied by human figures with plausible background content. We employ E$^2$FGVI (Li et al., 2022b), an optical flow-based inpainting method. E$^2$FGVI leverages temporal coherence and spatial context to generate high-quality inpainted frames, effectively removing humans while maintaining the integrity of the background.

After removing humans from the video frames, we focus on reconstructing human motion using 3D mesh models. The body mesh recovery module processes the resized videos to extract detailed 3D meshes of the human figures. We use the SMPL-X model for its comprehensive representation of the body, hands, and facial expressions, and the OSX algorithm for mesh recovery due to its multi-human support. The 3D meshes are then superimposed onto the inpainted frames, replacing real humans while retaining essential motion cues for accurate action recognition and anonymization. We also address occlusion challenges (Appendix B.2: Figure 7) involving peripheral objects in the scene. The augmentation framework (Figure 2) preserves objects typically occluded (on-body and peripheral) by the recovered mesh while maintaining privacy, handling erroneous occlusions and object integrity in pose estimation as follows:

**(1)** Segment the input image to obtain masks for all objects, including the human subject.

**(2)** Use the human mask to inpaint and remove the subject, creating a clean background.

**(3)** Do mesh recovery on the original image and overlay the mesh onto the inpainted background.

**(4)** Extract objects from the original image using their masks and overlay the composite image.

---

[1]We compute the relative entropy (Guo et al., 2016) between adjacent frames and select a frame from a subset with the least change in entropy. However, Kinetics consists of short videos showcasing a single action class, resulting in minimal entropy change between adjacent frames. Hence, selecting a random frame (from the mid-quartile range) per video is justified. The performance difference between the two methods was $< 0.1\%$.

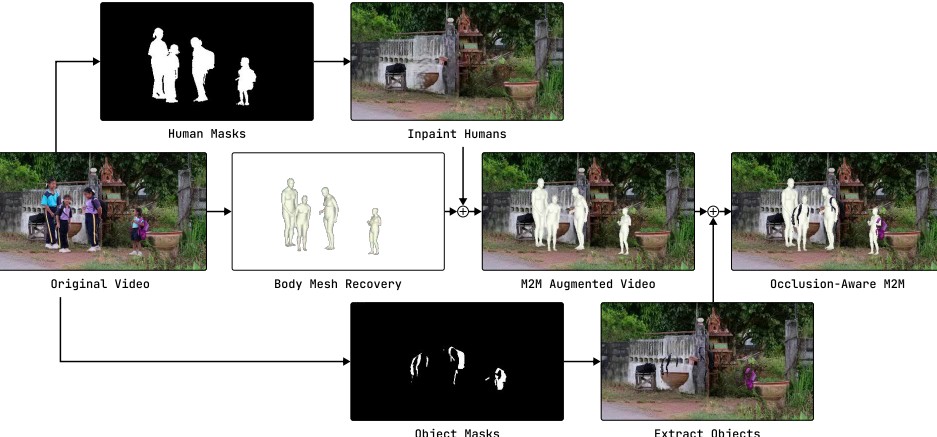

Figure 2: **MASK2MESH (M2M) Augmentation**. A visual representation of transforming real human actions into 3D mesh models across various activities for enhanced privacy and bias mitigation in action recognition. The figure details the flow from the initial video data of the Kinetics dataset, through masking, inpainting, and body mesh recovery to the final stage of mesh superimposition.

## 3.4 TRAINING PROCEDURE

Pre-training is vital in video action recognition, enabling models to learn generalized spatial and temporal features from large, diverse datasets. This foundation enhances the model's ability to recognize complex actions with less labeled data and improves performance and efficiency in downstream tasks (Schiappa et al., 2023). Our training procedure consists of two stages: (1) self-supervised pre-training utilizing VideoMAE; (2) supervised pre-training to ensure label alignment.

**Step 1: Video Masked Autoencoder for Self-Supervised Pre-training.** We employ the traditional training methodology for MAEs (He et al., 2021) tailored explicitly for video data. This entails a configuration consisting of an encoder and a decoder, in which the model acquires the ability to approximate masked pixel values within video frames. During this stage, the encoder and decoder undergo joint training. Once the model is sufficiently trained, the decoder is removed, leaving only the encoder.

**Step 2: Supervised Pre-training for Alignment of Labels.** The pre-trained VideoMAE encoder is then augmented with a linear classification head. The encoder and the linear classifier are trained in tandem, using the action labels for supervision.

**Putting it All Together.** M2M serves as the basis of our data preparation process across both the pretraining and alignment phases. During pretraining, M2M-augmented videos replace the original dataset, ensuring privacy-preserving inputs to the VideoMAE framework. This augmentation involves detecting human subjects, masking them, inpainting the occluded regions, and superimposing meshes. The resulting anonymized dataset allows VideoMAE to learn spatiotemporal representations in a self-supervised manner. During alignment, the same M2M pipeline is employed to prepare training data, ensuring the VideoMAE encoder aligns with action recognition labels. This two-step process ensures that the M2M augmentation is central to both stages of training, as summarized in Table 1.

## 3.5 DOWNSTREAM EVALUATION

The evaluation of our SMPLy Private models is conducted on six distinct action-recognition tasks. The UCF101 dataset (Soomro et al., 2012) comprises 13,320 YouTube videos spanning 101 action classes, showcasing notable diversity in activities performed and camera movement. The HMDB51 (Kuehne et al., 2011) dataset consists of 6,849 movie clips, categorized into 51 distinct action classes. The Diving48 dataset (Li et al., 2018) is a highly specialized data collection designed for competition diving. It consists of 18,000 video clips that cover 48 different categories. This dataset aims to evaluate our models' ability to handle the challenges posed by the identical background and object properties commonly found in competitive diving scenarios. Ikea Furniture (Han et al., 2017; Toyer

et al., 2017) Assembly provides a collection of 111 movies with 14 actors demonstrating assembling and disassembling furniture. These videos are filmed using the same camera and scenario settings, ensuring consistency. The videos are categorized into 12 different action categories. The UAV-Human dataset (Li et al., 2021) comprises 22,476 films recorded by unmanned aerial vehicles, such as drones, featuring 155 distinct action categories and 119 individuals. Note that from UCF101 to UAV-Human, the scene-object bias generally decreases.

# 4 EXPERIMENTS

For technical specifications and training information related to our experiments, see Appendix A. Note: SMPLy Private refers to our complete end-to-end pipeline, encompassing segmentation, mesh recovery, inpainting, VideoMAE pre-training, alignment, and downstream evaluation. M2M denotes the data augmentation method used to generate the meshed dataset (M2M Kinetics).

## 4.1 MASK2MESH PERFORMANCE

Table 1: **M2M Performance Evaluation.** Top-1 downstream task accuracy for linear probing (LP) and finetuning (FT) is reported. The mean FT and LP accuracy over all downstream tasks across datasets is represented in the final column. The results show that our SMPLy Private model outperforms, on average, prior benchmarks by at least 0.7% in FT and 0.1% in LP. If K-NEXUS is used (in teal), the improvement increases to at least 2.8% in FT and 1.4% in LP due to enhanced feature learning from class discretization. SMPLy Private rivals the VideoMAE trained with real human data (baseline in violet), reducing the realism gap by 0.5% in FT and 0.2% in LP. With K-NEXUS, SMPLy Private surpasses this baseline. All other scores are from (Kim et al., 2022; Zhong et al., 2023). Our choice of alignment was selected based on results from Table 4.

| Pre-trained Model | Privacy | Step 1: MAE | Step 2: Align | UCF101 | | HMDB51 | | Diving48 | | IkeaFA | | UAV-Human | | Mean | |
|---|---|---|---|---|---|---|---|---|---|---|---|---|---|---|---|
| | | | | FT | LP | FT | LP | FT | LP | FT | LP | FT | LP | FT | LP |
| VideoMAE-Align w/ Real | ✗ | Kinetics | Kinetics | 93.4 | 91.5 | 73.5 | 69.8 | 66.3 | 19.9 | 72.2 | 58.4 | 34.8 | 13.8 | 68.0 | 50.7 |
| TimeSformer w/ Kinetics | ✗ | - | Kinetics | 92.1 | 89.4 | 59.5 | 55.4 | 46.4 | 17 | 61.9 | 47.7 | 23.3 | 8.4 | 56.6 | 43.6 |
| TimeSformer w/ Synthetic | ✗ | - | Synthetic | 89 | 82.1 | 54.4 | 49.2 | 44.9 | 19.2 | 63.6 | 45.5 | 25 | 13.8 | 55.4 | 42.0 |
| TSN w/ RN50 | ✓ | - | Synthetic | 83.4 | 28 | 54.4 | 20.9 | 63.5 | 10.9 | 42.7 | 36 | 35.6 | 5.7 | 55.9 | 20.3 |
| I3D w/ RN50 | ✓ | - | Synthetic | 82.1 | 27.6 | 55.7 | 22.6 | 55.0 | 10.1 | 42.7 | 33.2 | 35.1 | 5.8 | 54.2 | 19.9 |
| OmniMAE-Align w/ Synthetic | ✓ | Synthetic | Synthetic | 80 | 26.4 | 53.3 | 22.2 | 57.3 | 10 | 41.5 | 35.7 | 31.8 | 5.5 | 52.8 | 20.0 |
| PPMA | ✓ | No Human Kinetics | No Human Kinetics + Synthetic | 92.5 | 88.4 | 71.2 | 64.9 | 64.0 | **21.9** | 67.9 | 57.7 | **38.5** | **19.3** | 66.8 | 50.4 |
| **SMPLy Priv. (Ours)** | ✓ | M2M Kinetics | M2M Kinetics | 93.2 | 90.9 | 72.6 | 69.2 | 66.0 | 19.7 | 71.3 | 58.2 | 34.6 | 14.3 | 67.5 | 50.5 |
| **SMPLy Priv. w/ K-NEXUS (Ours)** | ✓ | M2M Kinetics | M2M Kinetics | **94.2** | **91.6** | **74.3** | **70.8** | **69.0** | 21.6 | **72.9** | **59.5** | 36.4 | 15.3 | **69.6** | **51.8** |

Table 1 shows the average downstream performance on various classification tasks[2] with multiple pre-trained models, including our proposed SMPLy Private[3] and SMPLy Private with K-NEXUS. Models pre-trained on conventional large-scale real video data with humans typically have a performance edge over models trained with synthetic data due to a realism gap. We establish such a baseline by pre-training and then aligning VideoMAE with Kinetics-150 (first row in violet). Other privacy-preserving baselines present a significant realism gap (non-bolded and non-colored). However, with SMPLy Private and the use of our M2M-augmented dataset (M2M Kinetics), the average downstream performance gap from the human baseline is reduced to 0.5% with FT and 0.2% with LP.

The performance gap is attributed to SMPLy Private performing slightly worse than the human baseline on tasks with high scene-object bias, such as UCF101 and HMDB51. This is likely because the inclusion of humans in the Kinetics videos helps the model better learn both scene-object cues and action features. Compared to the performance of "OmniMAE-Align w/ Synthetic," SMPLy Private narrows the realism gap as it achieves a level of performance comparable to the "VideoMAE-Align w/ Real" baseline. Both "TSN with RN50" and "PPMA" utilize synthetic data for model training, yet they fall short of "VideoMAE-Align with Real" in downstream performance. With K-NEXUS,

---

[2]For fine-grained classification results, refer to Appendix C.

[3]Without K-NEXUS we use the Kinetics-150 classes as outlined by Zhong et al. (2023).

SMPLy Private further improves over "VideoMAE-Align w/ Synthetic" by 1.6% with FT and 1.1% with LP.

While Table 1 also demonstrates the impact of M2M across diverse downstream tasks, highlighting its integration into both pretraining and alignment stages, our approach outperforms PPMA in general. This showcases SMPLy Private's ability to reduce the performance gap with real-human pretraining. The results underscore M2M's contributions in bridging the realism gap, as evidenced by the improved accuracy in high scene-object bias tasks such as UCF101 and HMDB51. This improvement stems directly from the M2M augmentation, which provides a more robust representation of human actions during both MAE pretraining and supervised alignment. By contrast, "PPMA" relies on synthetic data and human-removed videos independently, limiting its ability to fully capture temporal dynamics.

Overall, SMPLy Private with K-NEXUS, which uses M2M Kinetics in both pre-training and alignment steps, achieves the best performance among privacy-preserving models, reducing the performance gap with the human-baseline model to minimal levels and even surpassing it. This shows the effectiveness of our approach in achieving high performance through pre-trained representations for privacy-preserving action recognition without the need for synthetic data.

## 4.2 AUGMENTATION STRATEGIES

The SMPL-X model provides a more holistic representation of the person's body than most segmentation strategies. This is due to the inherent property of SMPL-X to fit an entire human mesh to the bodies visible in the videos while preserving structural components. The segmentation methods are only responsible for laying a monocolored film outlining a person without any structural considerations. Unlike using only segmentation, the preservation of fine-grained details in the structure ensures that hand and finger movements remain distinct and traceable, even when they are close to or overlapping with the body (Appendix B.2: Figure 5). The depth information provided by the 3D nature of the mesh allows for a better understanding of spatial relationships between body parts, enhancing the overall representation. Hence, the $\approx 4\%$ improvement from solely using SAM segmentation as seen in Table 2. We incorporate the $E^2FGVI$ inpainting technique to remove humans

Table 2: **Analysis of Methods**. This table compares the impact of different combinations of methods (Row 1: SAM segmentation, Row 2: OSX mesh recovery, Row 3: $E^2FGVI$ inpainting) on privacy and downstream accuracy across various datasets.

| Segmentation | Mesh Recovery | Inpainting | Privacy | Downstream Accuracy (LP Only) | | | | | |
|:---:|:---:|:---:|:---:|:---:|:---:|:---:|:---:|:---:|:---:|
| | | | | UCF101 | HMDB51 | Diving48 | IkeaFA | UAV-Human | Mean |
| ✓ | ✗ | ✗ | ✓ | 86.6 | 61.4 | 18.9 | 53.9 | 13.0 | 46.8 |
| ✗ | ✓ | ✗ | ✗ | 91.1 | 69.4 | 20.0 | 58.3 | 14.6 | 50.7 |
| ✓ | ✓ | ✓ | ✓ | 90.9 | 69.2 | 19.7 | 58.2 | 14.3 | 50.5 |

from video streams. While direct application of mesh recovery to videos is feasible, integrating an inpainting step markedly improves the privacy-preserving capabilities of our framework. Absent this inpainting process, residual demographic information can still be discerned, compromising both the privacy and the unbiased nature of our framework (Appendix B.2: Figure 6). Although omitting inpainting yields a higher performance (by only 0.2%), applying inpainting ensures that our pipeline remains fully privacy-preserving. We hypothesize that the observed higher performance may stem from the retention of features, which, although insufficiently masked, provide additional discriminative features that aid the learning process of the model. This highlights a trade-off between performance and privacy – yet in this case, the privacy gains significantly outweigh the minuscule performance gains. As an aside, note that in a resource-constrained setting, OpenCV's implementation of Navier-Stokes inpainting (Bertalmio et al., 2001; Itseez, 2015) can be used. As inpainting is an intermediate step, any performance loss would be minimal.

Table 3: **Gender-Action Bias Analysis**. We show the performance on gender-biased tasks using real human data vs. 3D meshes. Neutral meshes achieve the highest average accuracy, demonstrating effective mitigation of gender-action bias.

| Method | Women in Men-Biased (FT) | Men in Women-Biased (FT) | Mean |
|---|---|---|---|
| VideoMAE w/ real humans | 81.4 | 78.8 | 80.1 |
| SMPLy Private w/ male meshes | 82.3 | **83.3** | 82.8 |
| SMPLy Private w/ female meshes | **83.4** | 82.5 | 82.9 |
| SMPLy Private w/ neutral meshes | 83.2 | 83.1 | **83.1** |

### 4.3 TO GENDER OR NOT TO GENDER?

In this section, we analyze the impact of using 3D meshes on gender-action bias in action recognition tasks[4]. Specifically, we compare the performance of a model trained on real human data from Kinetics-150. with those trained on our augmented meshed data (M2M Kinetics). We conduct experiments on a specifically curated split of the Kinetics dataset in which women perform male-dominated tasks and men perform female-dominated tasks. The results are summarized in Table 3. Training on real human data revealed significant gender-action bias, with lower performance on "men in women-biased" tasks compared to "women in men-biased" tasks. In contrast, models trained on 3D meshes showed improved performance. Male meshes increased accuracy for "men in women-biased" tasks, while female meshes for "women in men-biased" tasks. Neutral meshes performed consistently well across both subsets. Overall, using 3D meshes outperformed the real human data approach, with higher average scores across all mesh-based methods. This indicates that 3D meshes help mitigate gender-action bias by offering a gender-agnostic representation.

### 4.4 THE TORTOISE & THE HARE

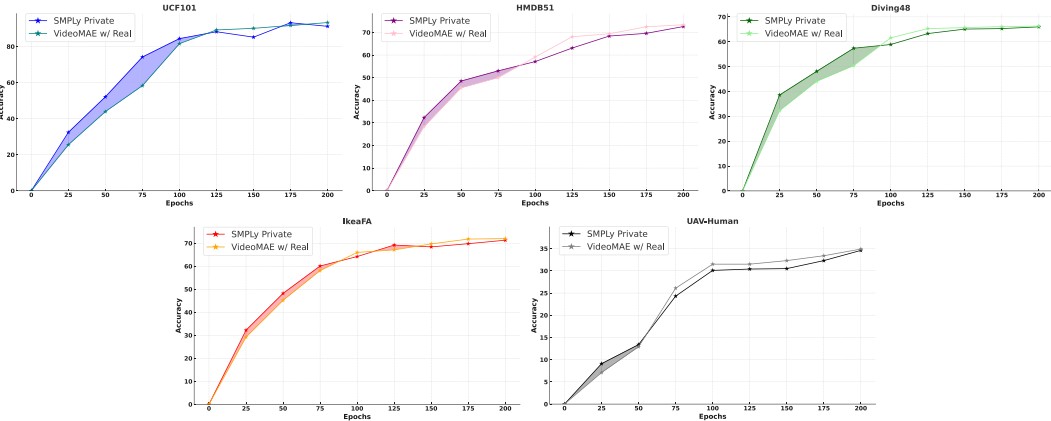

Figure 3: **Representation Learning Efficiency Comparison.** In the initial stages of training, the model trained with M2M Kinetics (teal in Table 1) demonstrates faster representation learning compared to the model trained on Kinetics (violet in Table 1), which catches up in the later stages.

Figure 3 demonstrates that our model, fine-tuned from each pre-training checkpoint on M2M-augmented data, learns representations quicker in earlier stages because humans are consistently depicted as meshes. This consistency allows the model to isolate and understand the actions performed by the meshes, whereas, in real data, the varied depictions of humans make representation learning more challenging. It is observed that our model learns representations faster for longer with higher scene-object bias as indicated by the descending size of the shaded areas from UCF101 to UAV-Human (left to right). This showcases another perspective on further closing the realism gap; however,

---

[4]See Appendix D for more details on this experiment. We used neutral meshes in all our main experiments.

the "tortoise" (VideoMAE trained on real human data) eventually catches and overtakes the "hare" (SMPLy Private), typically in the latter stages of training.

## 4.5 IN THE ALIGNMENT ARENA: HUMAN VS. MESH

Table 4: **Alignment Study for Humans and Meshes**. We present a comparative analysis of various alignment strategies following the pre-training of our model on the M2M Kinetics dataset. The alignment datasets under examination include the original Kinetics, Kinetics with humans removed, and M2M Kinetics. Our findings indicate that the M2M Kinetics dataset exhibits the smallest absolute difference, $|\Delta|$, from the Kinetics baseline (in violet), thereby further reducing the realism gap.

| Step 1: MAE | Step 2: Align | Privacy | UCF101 | | HMDB51 | | Diving48 | | IkeaFA | | UAV-Human | | Mean | | $|\Delta|$ from "real" | |
|---|---|---|---|---|---|---|---|---|---|---|---|---|---|---|---|---|
| | | | FT | LP | FT | LP | FT | LP | FT | LP | FT | LP | FT | LP | FT | LP |
| No Human Kinetics | Synthetic | ✓ | 91.4 | 81.9 | 71.5 | 62.0 | 65.3 | 21.8 | 67.3 | 57.7 | 38.3 | 20.8 | 66.7 | 46.0 | 1.1 | 4.7 |
| M2M Kinetics | No Human Kinetics | ✓ | 91.5 | 87.3 | 68.5 | 61.2 | 63.4 | 18.4 | 69.9 | 51.5 | 32.3 | 12.2 | 65.1 | 46.1 | 2.7 | 4.6 |
| | Kinetics | ✗ | **93.7** | **92.1** | **73.1** | **69.4** | **66.2** | **19.7** | **71.5** | **58.4** | 34.3 | 13.8 | **67.8** | **50.7** | 0.0 | 0.0 |
| | M2M Kinetics | ✓ | 93.2 | 90.9 | 72.6 | 69.2 | 66.0 | **19.7** | 71.3 | 58.2 | **34.6** | **14.3** | 67.5 | 50.5 | 0.3 | 0.2 |

In Table 4, we examine the effects of various alignment techniques following pre-training with the M2M Kinetics dataset. Our findings indicate that when the dataset is exposed only to backgrounds (the no-human scenario, involving only inpainting without mesh recovery), performance is significantly lower compared to when actual humans are shown to the model. This approach notably falls short of the Kinetics baseline (in violet) when the model is exposed to real humans during the alignment phase. However, when our pre-trained model is exposed to the M2M Kinetics dataset (with both inpainting and mesh recovery), we approximate the performance when compared to the Kinetics baseline ($|\Delta|$ is 0.3% and 0.2% for FT and LP respectively). This demonstrates that the mesh recovery technique in videos is effective in understanding real human actions, thus bridging the realism gap.

## 5 CONCLUSION

This study presents M2M, a framework that replaces humans in videos with detailed 3D SMPL-X meshes, ensuring privacy while maintaining action recognition accuracy. By augmenting real data, M2M avoids synthetic biases Kim et al. (2022). Models trained on M2M-augmented data outperform privacy-preserving benchmarks and rival non-privacy-preserving ones. A gender representation analysis shows gender-neutral meshes improve performance in biased classes. Additionally, M2M-augmented data accelerates representation learning, bridging the realism gap and standardizing action recognition datasets.

**Limitations & Future Work.** We perform image-instance segmentation without considering temporal relationships between frames. However, given that our data is video-based, an optimal approach would involve video-instance segmentation. This would likely yield more accurate masks and improved performance, mitigating issues such as transient mesh disappearances. Lastly, our present method focuses on body mesh recovery for individual video frames, whereas this process could be extended to entire videos to incorporate temporal relations. Most contemporary approaches to body mesh recovery leverage 3D pose estimation from videos, a complex problem (Needham et al., 2021). Accurately recovering 3D meshes in videos is challenging but could significantly reduce glitches and enhance performance. We can better capture and use temporal information by replacing frame-by-frame processing with video-level analysis. Lastly, we acknowledge the potential of the MANO model (Romero et al., 2017), which is tailored for hand-specific tasks in egocentric datasets such as Something-Something (Goyal et al., 2017) and EPIC-KITCHENS (Damen et al., 2018). These datasets primarily involve hand and wrist actions, presenting opportunities for further exploration.

**Ethical Implications.** By replacing real individuals in video footage with 3D meshes, M2M addresses privacy concerns and aligns with regulations like GDPR (Union, 2016) and ADPPA (Congress, 2022). It mitigates risks of identity theft and privacy invasion, while reducing biases related to race and gender. However, its capability to generate highly realistic video data could also be repurposed for more invasive surveillance systems, enhancing monitoring capabilities in workplaces or public areas and infringing on individual privacy and autonomy.

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

APPENDIX

# A    TECHNICAL SPECIFICATIONS

**General Details.** All our model training is distributed over NVIDIA 8×A100-80GB (SXM). In all experiments, we employ the ViT-B backbone and to uphold rigorous privacy preservation, we develop conduct training without pre-trained weights from ImageNet. We pre-train VideoMAE for 200 epochs using a tube masking technique that masks 90% of image patches to enhance learned video representations. After pre-training, we remove the VideoMAE decoder, retaining only the encoder. For label alignment, we conduct supervised pre-training for 50 epochs using the same subset of 150 Kinetics classes as SynAPT (Kim et al., 2022). For downstream evaluation, we fine-tune (FT) the entire network or train a linear probe (LP) for 30 epochs. Both steps use video inputs as 4D tensors $(C, T, H, W)$, with $C = 3$ (RGB channels), $T = 16$ frames, and spatial dimensions $H$ and $W$ as the video input is resized to $224 \times 224$ and normalized.

Table 5: **Summary of Training Details**.

| General Specifications | |
|---|---|
| GPU Configuration | NVIDIA 8×A100-80GB (SXM) |
| Model Backbone | ViT-B (12 encoder blocks, 768 emb. dim.; 4 decoder blocks, 384 emb. dim.) |
| Input Tensor Shape | $3 \times 16 \times 224 \times 224$ |
| Normalization | $\mu = [0.485, 0.456, 0.406], \sigma = [0.229, 0.224, 0.225]$ |
| **Self-Supervised Pre-Training** | |
| Pre-training Method | VideoMAE |
| Epochs | 200 (10 warm-up) |
| Masking Strategy | Tube, ratio = 0.9 |
| Batch Size | 128 |
| Patch Size | $2 \times 16 \times 16$ |
| Loss Function | MSE |
| Optimizer | AdamW |
| Learning Rate (Max) | 0.0008 |
| Learning Rate Scheduler | Cosine |
| **Supervised Label Alignment** | |
| Epochs | 50 (6 warm-up) |
| Loss Function | Cross-Entropy |
| Optimizer | AdamW |
| Learning Rate (Max) | 0.002 |
| **Downstream Evaluation** | |
| Adjustment Epochs (FT or LP) | 30 |

**Step 1.** During self-supervised pre-training, we use mean squared error (MSE) pixel reconstruction loss between the original and reconstructed frames. The batch size is 128, and patch sizes are $2 \times 16 \times 16$. We use the AdamW optimizer (Loshchilov & Hutter, 2019) with a cosine learning rate scheduler (Loshchilov & Hutter, 2017), a maximum learning rate of 0.0008, and a 10-epoch warm-up.

**Step 2.** For supervised label alignment, we add a final linear head to the ViT-B model for supervised training. The model shares an encoder with distinct linear classifiers for each dataset, using cross-entropy loss to measure discrepancies between the predicted and actual action categories. We train the model for 50 epochs with the AdamW optimizer and a cosine rate scheduler with a maximum learning rate of 0.002 and a 6-epoch warm-up.

# B DATASET AND METHODOLOGY CONSIDERATIONS

## B.1 MODIFIED CLUSTERING OPERATION AND DEFINITIONS

The K-NEXUS clustering operation, $\kappa(\bar{E}, K)$, aims to minimize the within-cluster bias and pairwise similarity between class embeddings. The clustering operation seeks to minimize the following objective, adapted from (Li et al., 2018), which incorporates both bias and pairwise similarity:

$$
\arg\min_{\omega} \sum_{k=1}^{K} \left( \sum_{\substack{\bar{e}_i, \bar{e}_j \in \omega_k \\ i \neq j}} \mathcal{F}(\bar{e}_i, \bar{e}_j) + \sum_{\bar{e} \in \omega_k} \mathcal{B}(D, \bar{e}) \right)
$$

Here, $\mathcal{B}(D, \bar{e})$ is the bias measurement for a dataset $D$ using class embedding $\bar{e}$ and is defined as:

$$
\mathcal{B}(D, \bar{e}) = \ln(\mathcal{M}(D, \bar{e})) - \ln(\mathcal{M}_{\text{chance}})
$$

where $\mathcal{M}(D, \bar{e})$ represents the performance of the representation $\bar{e}$ on dataset $D$, and $\mathcal{M}_{\text{chance}}$ is the performance at the chance level, defined as:

$$
\mathcal{M}_{\text{chance}} = \min_{\bar{e}} \mathcal{M}(D, \bar{e})
$$

**Pairwise Similarity Calculation.** For each class embedding $\bar{e}_i$, we calculate the average pairwise similarity with all other class embeddings $\bar{e}_j$ (where $i \neq j$). Let $\mathcal{F}(\bar{e}_i, \bar{e}_j)$ represent the similarity function (e.g., cosine similarity or entropy) between embeddings $i$ and $j$. The average similarity for class $c_i$ is defined as:

$$
M_i = \frac{1}{C-1} \sum_{\substack{j=1 \\ j \neq i}}^{C} \mathcal{F}(\bar{e}_i, \bar{e}_j)
$$

where $C$ is the total number of classes. This value $M_i$ represents how similar class $c_i$ is to all other classes in the dataset.

**Adjusted Bias Calculation.** We adjust the bias $B_i$ for class $c_i$ by comparing the pairwise similarity $M_i$ to a baseline chance value $M_{\text{chance}}$. The adjusted bias is given by:

$$
B_i = \ln(M_i) - \ln(M_{\text{chance}})
$$

This bias $B_i$ accounts for the relationships between class $c_i$ and other classes, and it is used to refine the bias measurements in the clustering process.

**Centroid Calculation.** The centroid $\bar{e}_k$ of cluster $\omega_k$ is defined as the mean of the bias measurements of all points in $\omega_k$:

$$
\bar{e}_k = \frac{1}{|\omega_k|} \sum_{\bar{e} \in \omega_k} \mathcal{B}(D, \bar{e})
$$

**Clustering Steps.** This involves the following iterative steps:

1. **Assignment step.** Assign each point $\bar{e}_i$ to the cluster with the nearest centroid based on both the pairwise similarity and bias measurements:

$$
\omega_k^{(t+1)} = \left\{ \bar{e}_i : \mathcal{B}(D, \bar{e}_k^{(t)}) + \sum_{j \neq i} \mathcal{F}(\bar{e}_i, \bar{e}_j) \leq \mathcal{B}(D, \bar{e}_j^{(t)}) + \sum_{l \neq j} \mathcal{F}(\bar{e}_j, \bar{e}_l), \forall j = 1, 2, \ldots, K \right\}
$$

2. **Update step.** Calculate the new centroids for each cluster:

$$\bar{\mathbf{e}}_k^{(t+1)} = \frac{1}{|\omega_k^{(t+1)}|} \sum_{\bar{\mathbf{e}}_i \in \omega_k^{(t+1)}} \mathcal{B}(D, \bar{\mathbf{e}}_i)$$

Furthermore, the optimization problem to select a subset of classes from the original dataset, as laid out in (Li et al., 2018), presents an exponential time complexity of $\mathcal{O}(2^n)$. It is possible to converge to a solution for the selection of a small number of classes. However, it lacks feasibility for our case ($K = 150$ to obtain the Kinetics-150 dataset). Our K-NEXUS approach, converges while having the time complexity of $\mathcal{O}(n \cdot k \cdot t)$, where $n$ is the number of classes, $k$ is the number of clusters, $t$ is the number of update steps. Thus, we are able to perform class sampling for larger values with a linear time complexity.

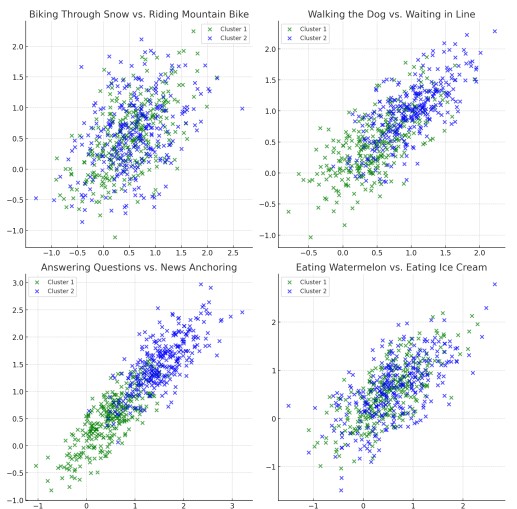

(a) **Example class clusters excluded by K-NEXUS.** The algorithm is designed to identify and exclude categories with overlapping visual cues or semantically broad definitions. It systematically excludes classes prone to ambiguity or high overlap within the embedding space (e.g., "answering questions" vs. "news anchoring" or "biking through snow" vs. "riding a mountain bike"). K-NEXUS considers these as "fine-grained" categories. Otherwise, visually distinct and contextually unique categories (e.g., "archery," "yoga") are retained.

(b) **Example class clusters included by K-NEXUS.** The algorithm is designed to identify and include categories with clearly separable visual cues or semantic definitions. It systematically includes classes that are easy to discretize within the embedding space (e.g., "yoga" vs. "archery" or "bowling" vs. "catching fish"). K-NEXUS considers these as "coarse-grained" categories. Otherwise, visually similar and contextually related categories (e.g., "eating watermelon," "eating ice cream") are discarded.

Figure 4: **Examples of class clusters identified and processed by K-NEXUS.**

In this work, we consider the K-NEXUS-selected classes as "coarse-grained" because they represent distinct, well-separated actions that rely less on subtle pose variations or fine contextual cues. These categories are designed to evaluate whether the our proposed framework can effectively learn high-level action semantics without relying on scene or background context. In contrast, "fine-grained" classes involve subtle distinctions, such as variations in hand positioning or object interactions, which present challenges even for fully-supervised models trained on real videos. For this reason, we classify the remaining 250 classes as "fine-grained." Our focus is not on hierarchical annotations or subtle interclass differences across datasets, but rather on the model's ability to handle categories with varying reliance on pose-level distinctions versus scene or temporal context. K-NEXUS classes are intentionally chosen to represent distinct, easily separable actions that primarily depend on human pose, making them coarse-grained for privacy-preserving mesh analysis. See Appendix C for more.

## B.2    SMPLY FAILING

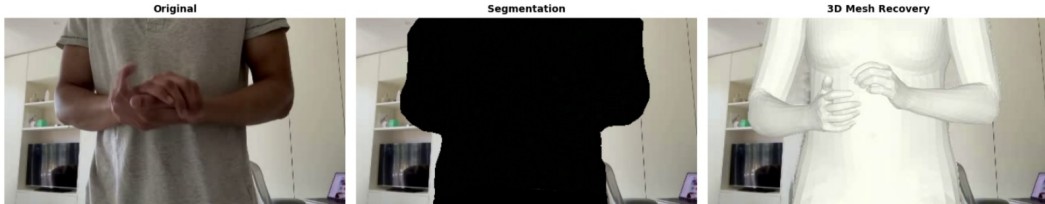

Figure 5: **Only Segmentation Fails.** The action of clapping is visually hidden when using only a segmentation mask but efficiently maintained using 3D mesh recovery post masking and inpainting.

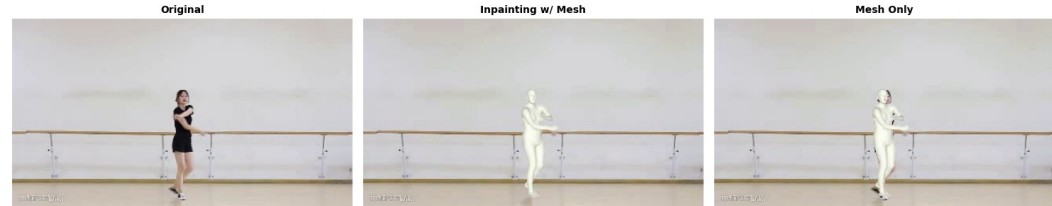

Figure 6: **Only Mesh Recovery Fails.** Employing only meshes exposes a significant chunk of the woman's face, leading to privacy leakage. SMPLy Private, which incorporates both inpainting and 3D meshes, preserves privacy.

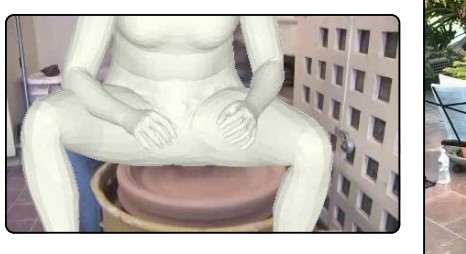 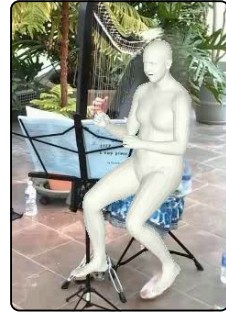 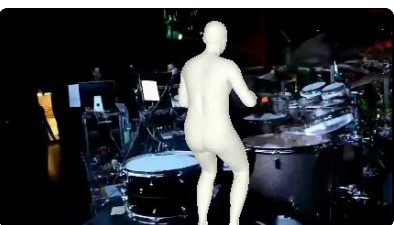

Figure 7: **Failure Cases.** SMPL mesh augmentation without M2M suffers when human joints are occluded. The pottery wheel, music stand, and drums are partially obscured by the superimposed mesh, demonstrating the challenges in handling occlusions within the scene.

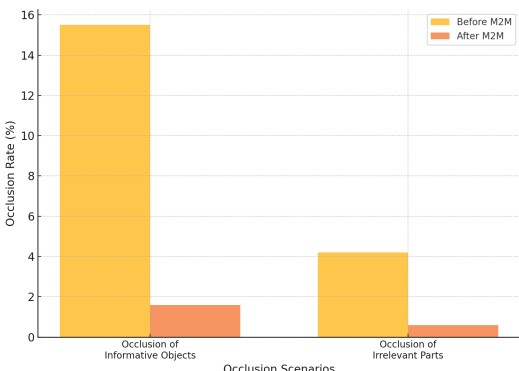

Figure 8: **MASK2MESH is occlusion aware.** In our investigation, we had challenges related to occlusions (Figure 7) using just SMPL meshes. To quantify this issue, we manually reviewed 20 randomly selected videos per class from Kinetics-150. Our findings indicated that occlusion-related difficulties were present in 15.5% of the videos. In these cases, the occlusions involved informative objects or backgrounds that contributed to learned features and supervisory signals. An additional 4.2% of cases also had occlusions; however, these did not involve the occlusion of significant objects or scenes essential to the video's labels (i.e., potentially irrelevant parts of the video were occluded, not the major components). Upon using the M2M occlusion-aware augmentation framework (Figure 2), both occlusion rates fell to 1.6% and 0.6% respectively (with 100% interrater reliability).

## C   FINE-GRAINED VS. COARSE-GRAINED ACTION CLASSIFICATION

Table 6: **Comparison of fine-grained and coarse-grained classification performance on Kinetics.** F-scores are reported as the average of mean cluster F-scores for fine-grained classification and as a simple mean F-score for coarse-grained classification. Refer to the end of Appendix B.1 for definitions on what is considered "fine-grained" and "coarse-grained" in this context.

| Method | Fine-grained | Coarse-grained |
|---|---|---|
| SMPLy Private w/ K-NEXUS | $53.3 \pm 0.12$ | **76.9** |
| SMPLy Private | **$69.5 \pm 0.08$** | 75.7 |

We specifically investigated the performance of our approach on both fine-grained and coarse-grained classification tasks. Given that K-NEXUS is designed to curate a set of classes that are coarse-grained, it was crucial to examine how well our model generalizes across fine-grained action classes within these broader categories. To this end, we clustered the Kinetics-400 classes into 150 coarse groups using K-NEXUS and then evaluated the accuracy within these clusters, where the classes are fine-grained. For the fine-tuning phase, we sampled 10% of these clusters (15 in total, filtering for clusters with only one class) and separately fine-tuned our core model on these selected clusters. We report the top-1 mean F-score to account for class imbalance, providing a more nuanced view of the model's performance.

We saw a noticeable performance drop in fine-grained classification when using K-NEXUS splits, with the F-score (53.3). This drop highlights the inherent challenge of fine-grained classification under the K-NEXUS framework. In contrast, when we repeated the experiments using a model pre-trained on random splits (Zhong et al., 2023), the fine-grained classification achieved a significantly higher F-score of 69.5. This suggests that while K-NEXUS is beneficial for coarse-grained classification (F-score > 75), it will introduce limitations for fine-grained tasks. However, it is important to note that for coarse-grained classification, the top-1 F-score on the Kinetics-400 test set for the 150 classes was greater when using K-NEXUS splits compared to random splits. This finding underscores the

strength of K-NEXUS in reducing class bias and enhancing generalization across diverse action categories, albeit with some trade-offs in fine-grained classification scenarios.

# D   ELABORATION ON GENDER STUDY

## D.1   DIFFERENCES BETWEEN MESH TYPES

The male, female, and neutral meshes differ primarily in their body shape and proportions, which are modeled to reflect biological and anatomical differences between the genders. The male and female meshes are gender-specific and trained on data tailored to their respective shapes, providing accurate body proportions and capturing gender-specific features like broader shoulders for males or wider hips for females. The neutral mesh is designed to be a compromise between male and female body shapes, enabling its use when the gender is unknown or ambiguous.

## D.2   HOW BIASED CLASSES WERE SELECTED

We chose women and male-biased classes based the default SMPL-X fitting method as it adapts to the human form within the real data. For instance, if a female human is seen in the video frame, the mesh overlayed will be of the female type. Upon manual inspection, during the process we undertook in Figure 8, we looked at the flipped and neutral cases too just by changing the gender parameter of the mesh for such mesh-fitted videos. Hence, this allowed us to easily categorize classes within the context of "Women in Men-Biased" or "Men in Women-Biased".

