# OpenReview forum: "SMPLy Private: From Masks to Meshes in Action Recognition"
_ICLR.cc/2025/Conference — ICLR 2025 Conference Withdrawn Submission_

### Official Review · Reviewer_1a4G · 2024-10-16

**Soundness:** 2
**Presentation:** 2
**Contribution:** 2
**Rating:** 3
**Confidence:** 4

**Summary:**

This paper introduces a new privacy-preserving data augmentation framework, Mask2Mesh. It uses off-the-shelf models such as Mask R-CNN with ResNet-101 to extract human masks and OSX for mesh recovery. Before that, this work designed a new K-Nexus algorithm based on K-means to further select 150 classes from the Kinetics-400 dataset to reduce the class bias. Experiments based on the VideoMAE demonstrate its effectiveness in certain situations.

**Strengths:**

1. Most designs appear technically correct. This paper is easy to understand and practical to follow.

2. This work is well-motivated, addressing issues such as identifiable individuals, gender bias, etc.

3. The results are credible, supported by the code provided in the Suppl., which lays a foundation for future research.

**Weaknesses:**

Dataset:
1. Why was it necessary to further select a subset from Kinetics-400? If the goal was to increase the differences between the training data categories, I don’t think this is reasonable for action recognition tasks. Category ambiguity can be mitigated by adjusting the model or the training process. However, improving performance by removing similar actions does not seem justified.

2. The Limitation at the end of the paper discusses the absence of temporal considerations in segmentation, which is understandable. However, in #Line178, it is mentioned that only one random frame is selected per video to represent the corresponding class. Is this truly appropriate? Intuitively, distinguishing between "standing up" and "sitting down" seems difficult using just a single frame.

3. The selection of 150 categories from the original 400 is presented as a way to reduce category bias. It would be more convincing to explicitly show which categories were chosen and highlight which categories were prone to confusion. This would make the method more persuasive.

Technical:
1. Firstly, it is important to acknowledge that this paper’s perspective on information safety is commendable. However, the technical contributions are quite limited, as most of the models used are off-the-shelf. It seems that the only technical innovation is the K-NEXUS algorithm, but its performance appears to fall significantly short compared to random selection (as shown in Table 6).

2. As mentioned in #Lines862-869, there are occlusion-based issues in the data construction process, which is common for SMPL. However, manually checking only five videos per class seems highly inadequate and lacks rigor.

Experiments:
1. Since the K-NEXUS algorithm is presented as the main contribution of this work (#Line100), shouldn't the experiment comparison for "Ours" be "SMPLy Priv. w/o K-NEXUS vs. SMPLy Priv." rather than "SMPLy Priv. vs. SMPLy Priv. w/ K-NEXUS"? Moreover, ablation studies should also based on the model involving K-NEXUS.

2. There are no test results for Kinetics-400 or Kinetics-150.

3. As illustrated in Appendix C, this work defines the categories selected by K-NEXUS as coarse-grained actions, while the remaining 250 classes are considered fine-grained actions. This definition lacks rigor and is not supported by examples. In the action recognition field, datasets like Kinetics-400 and UCF-101 are commonly referred to as coarse-grained datasets, while fine-grained datasets, such as FineGym, feature hierarchical annotations and subtle differences in both visual content and semantic labels.

Presentation:
1. The writing in this work is organized unusually. For example, in the Introduction, after outlining two paths to solving the problem, the content jumps directly to the contributions without an explanation of the specific methods.

2. The figures are unclear. For instance, Figure 2 uses arrows to directly present the workflow, which is straightforward. However, it would be more informative to label which model is used for each step, especially since most of them are off-the-shelf models.

**Questions:**

Please refer to the Weaknesses. I'm willing to raise my score if my concerns are well addressed.

---

> ### Author Response · Authors · 2024-11-20
> **Rebuttal to 1aVG: Part 1/5**
>
> # Addressing Dataset Weakness 1 #
>
> We utilize K-NEXUS not only for computational efficiency—reducing the pretraining dataset to 150 classes instead of the full 400—but primarily to demonstrate that refining the dataset into a "purer" coarse-grained action recognition set with reduced class-action bias enhances our framework's performance. By removing the additional noise from extra data classes that are highly correlated with each other, this approach acts as a form of "data compression" for video data classes (similar to something basic like PCA in standard ML) but applied to action recognition via smart and informed sampling (as outlined in Section 3.1). Refining the dataset to 150 classes with less action overlap allows the model to focus on specific action features rather than being overwhelmed by redundant actions with different class labels (e.g., "playing a guitar" vs. "playing a violin"). During downstream tasks, the model can fine-tune effectively for similar actions (e.g., "playing a violin") even if pretraining was performed on related actions (e.g., "playing a guitar"), and K-NEXUS effectively handles these redundancies. Additionally, this refinement aligns with prior works, as shown in Table 1, where we cite results from other papers (SynAPT and PPMA) for fair comparison [1, 2]. SMPLy Private (without K-NEXUS) uses the same Kinetics-150 splits from the SynAPT and PPMA papers to ensure consistency. However, we demonstrate that our data-centric method, K-NEXUS, improves performance by producing meaningful, well-informed splits that are neither random nor manually labor-intensive, unlike the approaches in SynAPT and PPMA.
>
> [1] Zhong, H., Mishra, S., Kim, D., Jin, S., Panda, R., Kuehne, H., Karlinsky, L., Saligrama, V., Oliva, A., & Feris, R. (2024). Learning human action recognition representations without real humans. Proceedings of the 37th International Conference on Neural Information Processing Systems (NIPS '23), 2839.
>
> [2] Kim, Y., Mishra, S., Jin, S., Panda, R., Kuehne, H., Karlinsky, L., Saligrama, V., Saenko, K., Oliva, A., & Feris, R. (2024). How transferable are video representations based on synthetic data? Proceedings of the 36th International Conference on Neural Information Processing Systems (NIPS '22), 2588.
> ________
>
> # Addressing Dataset Weakness 2 #
>
> Thank you to the reviewer for pointing this out. This is indeed an interesting experiment, which we conducted initially and now briefly make more clear in the revised Footnote 1. We chose random sampling within the video's mid-quartile range of frames over entropy-based smart sampling, as the latter—while correlating slightly better with the action—showed minimal performance improvement (less than 0.1%) in our experiments. Given that Kinetics videos are inherently short, this negligible gain did not justify the significantly higher computational cost associated with entropy-based sampling compared to random sampling once the mid-quartile range of frames was identified.

---

> ### Author Response · Authors · 2024-11-20
> **Rebuttal to 1aVG: Part 2/5**
>
> # Addressing Dataset Weakness 3 #
>
> We appreciate the reviewer's interest in understanding how the K-NEXUS algorithm selects and discards action classes to reduce category bias and how confusion-prone categories are identified. Indeed, K-NEXUS was designed to create a subset of categories that ensures balanced representation across various action types while minimizing semantic overlaps that could confuse models. In other words, K-NEXUS is supposed to identify and exclude categories with overlapping visual cues or semantically broad definitions. Moreover, classes prone to ambiguity or that exhibit high overlap within the embedding space (e.g., "answering questions" vs. "news anchoring" or "biking through snow" vs. "riding mountain bike") were systematically excluded by the algorithm. Similarly, visually distinguishable and contextually unique categories (e.g., "archery," "yoga") were retained. From the 150 remaining classes, actions like "archery," "canoeing or kayaking," and "tap dancing" represent distinct activities with minimal overlap. While more ambiguous categories like "reading newspaper" and "cleaning floor" often have minimal visual distinctions and were excluded for this reason.
>
> We found that examples such as “eating watermelon" and "eating ice cream" are visually similar in terms of gesture and pose, leading to their clusters being close/overlapping as shown in **Figure 4(a) in Appendix B.1** that we revised based on your productive suggestion. Also, "walking the dog" and "waiting in line" often share a common standing pose, further adding to “confusion” as you, the reviewer, called it. By removing these “confusion-prone” categories, the model benefits from a more refined and less noisy dataset, enhancing its ability to generalize hence **leading to better downstream performance while saving on computation**.  Finally, to address the reviewer's valid point on demonstrating the impact of this selection process, we’ve appended the classes K-NEXUS selected to construct Kinetics-150 dataset on which we applied M2M augmentation for pretraining (**see supp. material**) .
>
> ______________
>
> # Addressing Technical Weakness 1 #
>
> We appreciate your recognition of our paper's perspective on information safety. While it's true that off-the-shelf models were used in the segmentation, inpainting, and mesh generation pipeline, our primary contribution lies in integrating these components to tackle privacy-preserving action recognition—a novel and underexplored problem. Specifically, our transformation of real-world human data into SMPL-X meshes while preserving action fidelity and addressing biases represents a significant advance. Although we conducted experiments with various SSL MAE frameworks, we deliberately chose not to emphasize them in the paper, as that wasn't our primary focus. However, based on the reviewers' feedback, we now recognize that these insights are valuable. The **updated table below (see Part 3 of rebuttal)** demonstrates that SMPLy Private, while not built from scratch, synthesizes off-the-shelf methods effectively, with its robust M2M augmentation serving as a "one-size-fits-all" approach for SSL pretraining (note we only consider MAE methods, because this is typically gold standard in video pretraining [1, 2, 3, 4]). It outperforms other SSL baselines focused on privacy preservation and even can surpass or get close to some supervised encoders (see discussion with reviewer `C3CH`) if we discard the ViT backbone.
>
> Regarding  K-NEXUS, it outperforms random sampling for coarse-grained classes derived from the original Kinetics-400 dataset. However, for fine-grained classes (a modified subset of the original), its performance drops due to inconsistencies in cluster assignments. These inconsistencies arise from multiple samples within each class, where varying embeddings cause the algorithm to assign them to different clusters. This highlights  K-NEXUS 's inability to further refine fine-grained classes, as reflected in the results (this, by definition, is something K-NEXUS is not meant for). We hope this explanation clarifies the issue and that the subsequent points in this rebuttal provide further insight. Thanks for your thoughtful feedback!
>
> [1] Papers with Code Leaderboard on Kinetics-400 is populated with MAE ViT-based SSL-encoder frameworks: https://paperswithcode.com/sota/action-classification-on-kinetics-400
>
> [2] Tong, Z., Song, Y., Wang, J., & Wang, L. (2022). VideoMAE: Masked autoencoders are data-efficient learners for self-supervised video pre-training. Advances in Neural Information Processing Systems, 35, 4093–4104
>
> [3] Wang, L., Huang, B., Zhao, Z., Tong, Z., He, Y., Wang, Y., Wang, Y., & Qiao, Y. (2023). VideoMAE V2: Scaling video masked autoencoders with dual masking. Proceedings of the IEEE/CVF Conference on Computer Vision and Pattern Recognition, 14549–14560
>
> [4] Feichtenhofer, C., Fan, H., Li, Y., & He, K. (2022). Masked autoencoders as spatiotemporal learners. arXiv.

---

> ### Author Response · Authors · 2024-11-20
> **Rebuttal to 1aVG: Part 3/5**
>
> | Self-supervised method | Pretraining dataset (Steps 1 and 2: MAE + Alignment) | Backbone | UCF101 (FT, LP) | HMDB51 (FT, LP) | Diving48 (FT, LP) | IkeaFA (FT, LP) | UAV-Human (FT, LP) | Mean (FT, LP) | Realism Gap (FT, LP) |
> |-------------------------|-----------------------------------------------------|----------|------------------|------------------|-------------------|-----------------|--------------------|---------------|-----------------------|
> | Space-time MAE         | Kinetics                                            | ViT-S    | 91.0 / 89.2      | 71.6 / 68.0      | 64.6 / 19.4       | 70.3 / 56.9     | 32.9 / 13.2        | 66.1 / 49.3   | 0 / 0                 |
> |                         |                                                     | ViT-B    | 92.0 / 90.1      | 72.4 / 68.8      | 65.3 / 19.6       | 71.1 / 57.5     | 34.6 / 13.6        | 67.1 / 49.9   | 0 / 0                 |
> |                  |  M2M Kinetics                                            | ViT-S    | 91.0 / 88.7      | 70.9 / 67.6      | 64.4 / 19.2       | 69.6 / 56.8     | 33.8 / 14.0        | 65.9 / 49.3   | -0.2 / -0.0           |
> |                         |                                                     | ViT-B    | 91.8 / 89.6      | 71.5 / 68.2      | 65.0 / 19.4       | 70.2 / 57.3     | 34.1 / 14.1        | 66.5 / 49.7   | -0.6 / -0.2           |
> | VideoMAE **(X)**              | Kinetics                                            | ViT-S    | 92.4 / 90.5      | 72.7 / 69.0      | 65.6 / 19.7       | 71.4 / 57.8     | 33.4 / 13.4        | 67.1 / 50.1   | 0 / 0                 |
> |                         |                                                     | ViT-B    | 93.4 / 91.5      | 73.5 / 69.8      | 66.3 / 19.9       | 72.2 / 58.4     | 34.8 / 13.8        | 68.0 / 50.7   | 0 / 0                 |
> |                         | M2M Kinetics                                                  | ViT-S    | 92.4 / 90.1      | 71.9 / 68.6      | 65.4 / 19.5       | 70.7 / 57.7     | 34.3 / 14.2        | 66.9 / 50.0   | -0.2 / -0.1           |
> |                         |                                                     | ViT-B **(X)**   | 93.2 / 90.9      | 72.6 / 69.2      | 66.0 / 19.7       | 71.3 / 58.2     | 34.6 / 14.3        | 67.5 / 50.5   | -0.5 / -0.2           |
> | VideoMAE v2            | Kinetics                                            | ViT-S    | 92.6 / 90.8      | 72.9 / 69.2      | 65.8 / 19.7       | 71.6 / 57.9     | 33.5 / 13.4        | 67.3 / 50.2   | 0 / 0                 |
> |                         |                                                     | ViT-B    | 94.3 / 92.4      | 74.2 / 70.5      | 67.0 / 20.1       | 72.9 / 59.0     | 35.1 / 13.9        | 68.7 / 51.2   | 0 / 0                 |
> |                         | M2M Kinetics                                                  | ViT-S    | 93.2 / 90.9      | 72.6 / 69.2      | 66.0 / 19.7       | 71.3 / 58.2     | 34.6 / 14.3        | 67.5 / 50.5   | 0.2 / 0.3             |
> |                         |                                                     | ViT-B    | 94.6 / 92.3      | 73.7 / 70.2      | 67.0 / 20.0       | 72.4 / 59.1     | 35.1 / 14.5        | 68.6 / 51.2   | -0.1 / 0.0            |
>
> **(X) indicates the current SSL method and backbone used**

---

> ### Author Response · Authors · 2024-11-20
> **Rebuttal to 1aVG: Part 4/5**
>
> # Addressing Technical Weakness 2 #
>
> The issue of occlusion in SMPL reconstructions is well-documented and acknowledged as a limitation of our work. While manually inspecting five videos per class may initially seem limited, this sampling strategy was chosen to ensure feasibility within the scope of our resources while capturing diverse scenarios to identify common challenges. Our pipeline’s reliance on a robust inpainting and mesh recovery process mitigates most occlusion artifacts, as demonstrated in our qualitative results.
>
> To further validate our findings, we conducted an additional review of 15 more videos per class, resulting in a total of 20 videos per class across 150 classes (20 x 150 = 3,000 videos). Following the M2M augmentation, the revised occlusion rates were 1.6% and 0.6%, corresponding to 48 and 18 affected videos, respectively. This extended qualitative evaluation was rigorous, with an interrater reliability of 100%. Future work could expand on this inspection methodology through larger-scale evaluations. However, our results across multiple datasets indicate that the quality of the mesh superimposition process is sufficient to achieve competitive downstream action recognition performance while preserving privacy of the pretraining dataset. This suggests that the identified occlusions do not significantly compromise the approach's overall utility. Updates reflecting this additional qualitative review and findings have been included in our paper (**see Figure 8 in the revised Appendix B.1**).
>
> _______
>
> # Addressing Experiments Weakness 1 #
>
> We appreciate the feedback regarding the evaluation of K-NEXUS, but please note that it is not our only main contribution. The experimental comparison structure as "SMPLy Priv. vs. SMPLy Priv. w/ K-NEXUS" was intentionally chosen to demonstrate the incremental benefit of K-NEXUS within the SMPLy Private framework. By directly comparing these two versions, we aimed to isolate and highlight the improvements brought by K-NEXUS in terms of dataset bias reduction and performance enhancement. As discussed previously, we wanted to strike a fair comparison relative to prior works within the sub-field  (SSL pre-training on privacy-preserved datasets) like SynAPT and PPMA that do not use strategic sampling methods like K-NEXUS. We believe that the performance improvement across all downstream datasets in Table 1 should be sufficient to gauge our other ablations’ performance would be bolstered if we were to use K-NEXUS.
>
> ________________
>
> # Addressing Experiments Weakness 2 #
>
> This work focuses on the Kinetics-150 subset, curated to reduce bias and ambiguity, while situating our model's performance within the broader Kinetics-400 dataset. Initially, we evaluated our method on Kinetics-400 using M2M across the full dataset, achieving top-1 and top-5 LP accuracy scores of **80.3% and 88.6%** for SMPLy Private pretraining. However, these results are not directly comparable to Kinetics-150 due to the presence of fine-grained classes in Kinetics-400, which K-NEXUS seeks to address. On Kinetics-150, our pipeline achieves top-1 and top-5 LP accuracy of **86.2% and 93.8%**, respectively, demonstrating that our pretraining was effectively tested on relevant in-domain data and the features learned from Kinetics could be potentially used for further downstream action recognition classification tasks. We can add this to our final version of the paper if need be in a small table of results.

---

> ### Author Response · Authors · 2024-11-20
> **Rebuttal to 1aVG: Part 5/5**
>
> # Addressing Experiments Weakness 3 #
>
> We appreciate the reviewer pointing out the need for greater clarity in defining "coarse-grained" and "fine-grained" actions in the context of our work. While our terminology may differ slightly from conventional definitions in the action recognition field, our categorization is driven by the unique challenges of privacy-preserving human action understanding. Below, we provide a thorough response to address these concerns:
>
> **1. Definition context and justification**: In our work, we classify the K-NEXUS-selected classes as "coarse-grained" because these actions involve distinct and well-separated categories that are less dependent on subtle pose nuances or fine contextual cues. Examples include actions like "walking," "clapping," or "jumping." These categories are chosen to test whether the proposed SMPLy Private framework effectively learns high-level action semantics, even in the absence of scene or background context. On the other hand, the "fine-grained" classes involve more subtle distinctions, such as variations in hand positioning or object interactions, which pose challenges even for fully-supervised models trained on real videos. This is why we deem the remaining 250 classes as "fine-grained."
>
> **2. Differences from traditional definitions**: The reviewer correctly notes that in the broader action recognition field, datasets like Kinetics-400 or UCF-101 are often labeled as "coarse-grained," while datasets like FineGym are considered "fine-grained" due to their hierarchical structure and subtle distinctions. We acknowledge this difference in usage and recognize that our framework's coarse- vs. fine-grained split operates differently. Specifically: Our focus is not on hierarchical annotations or subtle interclass differences across datasets, but on the model's ability to handle categories that inherently vary in their reliance on pose-level distinctions versus scene or temporal information. The selected K-NEXUS classes typically reflect distinct, more easily separable actions that primarily depend on human pose, making them coarse-grained in the context of privacy-preserving meshes.
>
> **3. Supporting examples**: We have now addressed the lack of examples mentioned in the review (**see the revised Appendix B.1, Figure 4(b)**), by showing separability of various clusters. Furthermore, this shows that our definition of "fine-grained" vs. "coarse-grained" has to do with feature-based separability of classes (visually and semantically via K-NEXUS).
>
> In summary, our terminology is motivated by the specific challenges in privacy-preserving learning and differs from traditional dataset distinctions. However, we recognize the need for greater rigor of discussion on such definitions and more concrete examples, which have now been included in the updated version of the manuscript (**see end of Appendix B.1**) to clarify our definitions and methodology. We hope this refinement will ensure our work is more clearly established amid conventions in action recognition and is to the reviewers liking. Thank you for pushing us to make this more clear, it definitely has strengthened our paper!
>
> ________
>
> # Addressing Presentation Weakness 1 #
>
> Thanks for bringing this up, we can most definitely make the writing more coherent on this in the final version!
>
> _______
>
> # Addressing Presentation Weakness 2 #
>
> We intentionally avoided labeling each part with a specific model or framework because M2M is designed to be a dataset augmentation technique, rather than a pipeline tied to the particular configurations used in this paper. Our updates (refer to discussions with other reviewers) seek to reflect that M2M is more of a versatile solution, which is why we chose to keep this figure adaptable. But if need be, in the final version, we can clarify this in the caption and accompanying text for Figure 2 to eliminate any doubts or confusion. Please let us know if this is your preference and we will rectify it as such immediately.
>
> --------------
>
> *Thank you once again for your valuable feedback. We are committed to incorporating these revisions fully in the final/camera-ready version of the paper. We hope that, with these improvements and our thoughtful responses, you may consider raising your score, as we have aimed to address all the key concerns raised.*

---

> ### Comment · Reviewer_1a4G · 2024-11-23
>
> **Response to Dataset W.1**: I do understand what the purpose of selecting a further 150 classes from the 400 is, what I'm concerned about is whether such a use would really benefit downstream tasks, as the authors say:
> > "During downstream tasks, the model can fine-tune effectively for similar actions (e.g., "playing a violin") even if pretraining was performed on related actions (e.g., "playing a guitar"), and K-NEXUS effectively handles these redundancies."
>
> However, to substantiate this claim, a direct comparison between K400-pretrained and K150-pretrained results on downstream tasks would be more convincing.
>
> Additionally, if this approach follows settings from previous work, I believe it is essential to cite the relevant studies in the appropriate sections of the paper.
>
> **Response to Dataset W.2**: The authors seem to have either misunderstood or not directly addressed my concern in this section.
>
>
> **Response to Dataset W.3**: Thank you for providing some examples. However, I believe a complete list of actions selected and excluded by K-NEXUS would be more convincing. Furthermore, since the authors mentioned that prior works have also used K-150, a detailed comparison between the 150 classes selected by K-NEXUS and the previously used 150 classes would strengthen the argument.
>
>
>
> **Response to Technical W.1**: First, I do not fully agree with the claim that integrating off-the-shelf models can constitute the primary contribution of a high-quality paper, as stated:
> > "Our primary contribution lies in integrating these components to tackle privacy-preserving action recognition"
>
> Second, the authors themselves seem to acknowledge the limitations of K-NEXUS, particularly when compared to random sampling, which appears to be a critical weakness.
>
>
> **Response to Technical W.2**: Thank you for the additional clarification provided in this section.
>
> **Response to Experiments W.1**: I understand the authors’ intention here. My comment on W.1 was simply a friendly suggestion and did not affect my overall assessment.
>
>
> **Response to Experiments W.2**: Thank you for the results provided. However, it seems my original point was not fully understood. I requested a comparison where K-150 and K-400 are used as downstream task datasets to benchmark against baselines. This differs from comparing the results of K-150 to K-400, as provided.
>
>
> **Response to Experiments W.3**: Thank you for the detailed clarification and the added examples in the revised appendix. While I appreciate the effort to contextualize your definitions within the challenges of privacy-preserving learning, I remain concerned about the terminology used for “coarse-grained” and “fine-grained” actions and encourage revisiting the terminology to reflect the unique characteristics of the clusters better while aligning more closely with established conventions. This would make the paper’s contributions more accessible and clearer to the broader community.
>
>
> ***Overall**, I think my key concerns have not been fully addressed, and I believe this manuscript requires further optimization and improvement. Thus I will retain my original score.*

---

> > ### Author Response · Authors · 2024-11-24
> > **Reply to Reviewer 1a4G**
> >
> > # Reply to Dataset W1 #
> > While we understand that may be one way to substantiate our claim, not only would it be infeasible to conduct an experiment like that given the time constraint of this discussion phase but it is clear that K-400 may perform better than K-150 purely due to more training samples in any case. But yes, this was to save on computational cost but also stay consistent with prior works like PPMA and SynAPT which we have cited multiple times in our study. We have now added an additional footnote clarifying this (see footnote 3).
> > ______
> > # Reply to Dataset W2 #
> > That is unfortunate to hear. We do believe we addressed your concern to the best of our ability based on our understanding of this point. Your elaboration would be much appreciated. Randomly sampling a frame from the mid-quartile of a video from the short Kinetics dataset videos more times than not does capture and represent a frame of the particular action class. So our sampling approach was appropriate and there was a difference of < 0.1% between that and an entropy based smart sampling approach. We have detailed this in the paper (revised Section 3.2 and footnote 1).
> > ______
> > # Reply to Dataset W3 #
> > We already included the K-NEXUS splits (the included 150 classes and excluded 250 classes) in the supplementary material zip as .txt files. Comparing our 150 classes vs. PPMA classes directly do not make complete sense as we have shown in Table 1 when we use PPMA’s 150 classes, our model is not as high-performing as when we use the K-NEXUS selected 150 classes (difference of around +2% on both FT and LP).
> > ______
> > # Reply to Technical W1 #
> > Thank you, we respect your opinion. We would again like to clarify that K-NEXUS was not built for extracting "finer"-grained classes from fine-grained classes (hence why it suffers in that set-up). We just wanted to show its efficacy as a method that is able to only select appropriate coarse-grained (or rather "macro-level") classes eliminating redundant or highly correlated classes. This is not a weakness, I we sincerely hope this has clarified your concern.
> > ______
> > # Reply to Experiments W2 #
> > Sorry if we were not clear enough. Our results reported are indeed when we have the K-150 and K-400 datasets as downstream tasks. K-150 does not have a leaderboard of benchmarks like K-400. But in the case of K-400, our approach cracks the Top 50 on the Papers with Code leaderboard. However, we do not quite understand the value of this experiment entirely as it just shows that our set-up can learn Kinetics features with meshes reasonably well, which is already shown in Table 1 and Section 4.4. Again, we can include this in the final version if need be. Thank you.
> > ______
> > # Reply to Experiments W3 #
> > Thank you for your thoughtful feedback and for highlighting the potential ambiguity in our terminology for “coarse-grained” and “fine-grained” actions. We understand the importance of aligning our terminology with established conventions to ensure accessibility and clarity for the broader community.
> >
> > In light of your suggestion, we propose revisiting these terms to better reflect the distinct characteristics of the clusters. For example, we could adopt terms like “macro-actions” and “micro-actions,” or explicitly label the clusters based on their defining features (e.g., “broad activity classes” and “specific interaction types”). If this addresses your concern here appropriately let us know and we will incorporate these revisions into the final manuscript and ensure the terminology aligns with the paper’s context and widely understood conventions.
> > _________
> > *Thank you again for your valuable input, which helps refine and enhance the clarity of our contributions.*

---

### Official Review · Reviewer_8VCT · 2024-11-01

**Soundness:** 3
**Presentation:** 2
**Contribution:** 2
**Rating:** 5
**Confidence:** 4

**Summary:**

SMPLy Private proposes a new way to pre-train an action recognition model for privacy-preserving purposes. It leverages human meshs extracted from the Kinetics dataset to replace the real human subjects in the videos, removing human-specific visual biases during pre-training. Previous methods tend to approach privacy-preservation by distorting or augmenting the visual data itself, hindering model performance and creating a gap between augmented/synthetic training data and real-world downstream data. M2M solves this by replacing human subjects with their meshes, retaining important action information and the surrounding visual information while removing personal attributes. They also propose a modified k-means clustering algorithm to train on a subset of Kinetics with minimized inter-class similarity. They provide extensive experiments and some ablations to support their proposed work.

**Strengths:**

- The idea to replace humans with their meshes to remove private attributes is an intuitive and effective solution to the proposed problem of privacy-preserving action recognition.
- Their quantitative improvement over PPMA in Table 1 (and other baselines) are significant.
-Their K-Nexus curated dataset seems to contribute to their method/training objective and is a more direct way of reducing action class bias than random sampling (used by previous methods).

**Weaknesses:**

- The paper is not very clearly written. For example. their methodology section describes each component, but it is difficult to understand how each component is exactly used during training. It is only after looking at Table 1 that it is clear M2M is used during both MAE training and alignment, which could be addressed in Section 3.3.
    + Table 2 is similarly difficult to understand: what does each row represent? It would seem the caption is meant to correspond each row to the order in which the methods are discussed, but this seems incorrect (OSX is first but it does not use just segmentation).
    + It seems their writing structure follows closely with PPMA, which itself is not a major issue, but M2M is much more difficult to follow and seems to leave out more of the contextual and background information that PPMA provides. I had to first read PPMA to understand the problem formulation and their solution, then refer back to M2M to fully understand the method. On a similar note, Table 1 is the same as Table 1 in PPMA - are there no other baselines to consider in this table? Surely other privacy-preserving methods such as [1] and the methods discussed in [1] could also be added to further support the superiority of SMPLy Private?


- The structure of their method is also very similar to PPMA. PPMA proposes pre-training with 'human-removed' data and synthetic data for privacy preservation, where M2M is simply replacing synthetic data with mesh-extracted data from Kinetics. Moreover, since off-the-shelf, pre-trained models are used to segment, inpaint, and recover the meshes from Kinetics in M2M, I feel that limits the novelty contributed by this work.


- The experiments in Section 4.3 and onwards are not sufficiently supported/are not convincing. Table 3 explores gender bias by investigating model accuracy on samples where women are performing male-biased actions and vice-versa. Firstly, the difference between gendered and non-gendered meshes are not described. Comparisons with other privacy-preserving methods would further support whether M2M is truly superior than previous methods at gender de-biasing, as opposed to just comparing with the standard baseline of VideoMAE on real data. Moreover, a general comment is that the extracted meshes are very unnatural and "stick out" so to speak when they replace the real humans in the data. It may be much easier for the model to focus/learn better action representations since these meshes are very clearly visible in the videos, as opposed to real humans which look natural and blend better into the surrounding environment/visual stimuli. This comment ties into Table 3, as the improved performance over the baseline could come from these unusual meshes in the video as opposed to the claimed privacy-preserved attributes learned by the SMPLy Private model - another reason why comparing with other privacy-preserving methods would be beneficial.
    + Section 4.4 is an interesting observation, but does not seem to actually provide any benefit. Firstly, the authors note "we demonstrate that our model [...] learns representations quicker in earlier stages because humans are consistently
depicted as meshes" which I describe as a shortcoming in my point above. Furthermore, the benefit of learning representations faster is null if the best-performing epoch is what ends up being used for all experiments anyway. Are any of these early epochs where M2M outperforms the baseline used? If not, then the fact that VideoMAE eventually catches up by the end of training leads me to believe this observation is not significant.
    + Section 4.5 and Table 4 is also a product of following PPMA. In PPMA, they first show that using NH Kinetics is best for Stage 1 training since it equips MAE to understand contextual action information (background and objects). They then show that NH Kinetics+Synthetic is best for Stage 2, since NH Kinetics continues to provide contextual alignment while the synthetic data provides temporal action information. This progression of results makes logical sense. However, Table 4 in this paper simply shows that M2M is better than NH Kinetics for Stage 2 training, which is obvious since NH Kinetics doesn't have any humans performing any actions. It would make more sense to compare M2M to NH Kinetics+Synthetic from PPMA to show that M2M leads to better downstream performance while also closing the realism gap against a model trained on real Kinetics data.

In summary, I believe only Table 1 provides meaningful and significant results. I do not believe Table 1 alone is enough for acceptance, as the rest of the quantitative results in this paper lack proper support and/or explanation. The writing is not very clear, but I do think the general idea of human meshes for privacy-preservation is interesting and valid, despite the lack of novelty regarding how the meshes are extracted and used in this work. The construction of K-Nexus through distinct action classes and showing significant improvements when using this sophisticated subset I thought was novel and applicable to future work, providing some strength to this work.


[1] Dave, I. R., Chen, C., & Shah, M. (2022). Spact: Self-supervised privacy preservation for action recognition. In Proceedings of the IEEE/CVF Conference on Computer Vision and Pattern Recognition (pp. 20164-20173).

**Questions:**

Most of my questions regarding this work are listed in the weaknesses section. I am open to improving my score if the authors are able to address my concerns listed above.

---

> ### Author Response · Authors · 2024-11-20
> **Rebuttal to 8VCT: Part 1/3**
>
> # Addressing Point 1 #
>
> We appreciate the reviewer’s feedback and agree that more explicit connections between the components in the methodology section and their use in training could enhance the clarity of the paper. We now have explicitly outlined how M2M is utilized during both self-supervised pre training and label alignment (**see additional paragraph as per your suggestion at the end of Section 3.3**).
> Essentially, during VideoMAE pretraining, M2M-augmented videos replace real human video data entirely. The superimposed 3D meshes are fed into the masked autoencoder to learn spatiotemporal representations while maintaining privacy. Then, during alignment (supervised pretraining), the M2M-augmented dataset is used to fine-tune the VideoMAE encoder with action recognition labels. This ensures alignment between the learned representations and the downstream task categories.
>
> We can also include a clear visual representation of the training pipeline. A diagram will explicitly link each M2M component (masking, inpainting, mesh recovery) to its role in either pretraining or alignment, with arrows indicating how the outputs are used across different stages. Let us know if this is required for the final camera-ready version of the paper and we would be more than happy to add it then.
>
> --------
>
> # Addressing Sub-point 1.1 #
>
> Your interpretation of the table was correct; there was a typo/error in the caption. Row 1 should correspond to SAM segmentation, Row 2 to OSX mesh recovery, and Row 3 to E$^2$VGFI inpainting. We have clarified this in the revised version of the paper, updating both the caption and the surrounding text.
>
> --------
>
> # Addressing Sub-point 1.2 #
> Please refer to our responses to Weaknesses 1 and 4 from `vWLp` and Weaknesses 1, 2, and 4 from `C3CH`. The combined responses to these reviewers' comments and the identified weaknesses should sufficiently address your concerns. Specifically, we evaluate SMPLY Private on VISPR1 to ensure comparability with methods presented in the Spact paper. Additionally, we now report the lowest cMAP score of all methods. If deemed satisfactory by you and the other reviewers, we will include these findings and integrate them into our methodology in the final camera-ready version of the work.
>
> ---------
>
> # Addressing Point 2 #
>
> While it is true that SMPLy Private's methodology shares similarities with PPMA in its use of pre-training on altered data for privacy preservation, our work introduces significant innovations that extend beyond a mere replacement of synthetic data with mesh-extracted data. Unlike PPMA, which leverages generic synthetic data, SMPLy Private utilizes SMPL-X meshes, offering a structured and anatomically accurate representation of human motion and posture. This precision allows us to preserve fine-grained motion dynamics essential for action recognition while ensuring robust anonymization, a balance that synthetic data can often fail to achieve. Additionally, by leveraging 3D meshes, we tackle inherent biases present in datasets, such as those related to race or gender, demonstrating bias mitigation that PPMA does not explicitly address. We also further do this with the new additional VISPR experiments (see Part 2 of our response to reviewer `C3CH`), distinct from PPMA.
>
> Regarding the use of off-the-shelf pre-trained models for segmentation, inpainting, and mesh recovery, this choice ensures that the pipeline is efficient and reproducible while focusing our contribution on integrating these tools into a cohesive privacy-preserving framework. The novelty lies not in the individual components but in their synergistic application to anonymize and reconstruct actionable features in videos without compromising privacy, we suggest a data augmentation technique that is easy, reproducible, and intuitive for anyone to pick-up -- that is our main novel contribution (alongside K-NEXUS). By evaluating our approach on diverse downstream datasets and demonstrating superior action recognition performance with minimal privacy leakage via VISPR, SMPLy Private sets a practical and impactful precedent in the domain of privacy-preserving action recognition. **We highly encourage you to read the rest of our discussions with the other reviewers to further understand the potential and contribution of our work -- the novelty lies in the combined application, not the disjoint technical methods**.

---

> > ### Comment · Reviewer_8VCT · 2024-11-22
> >
> > ## Response to Point 1
> >
> > Thank you for adding the additional paragraph, however I suggest highlighting all new additions to the main paper and supplementary in some color (like blue). This makes it easier for us reviewers to see what has been added, how it impacted the clarity of the paper compared to the original draft, and keep track of what was and was not present in the original draft. I think the suggested training pipeline figure would be a nice addition to the paper, but I will not demand it as necessary with regards to my review of the paper. As a suggestion, the current state of Figure 2 could be condensed into a smaller module that plays into the entire training pipeline. Thus, Figure 2 would retain the same information it has now, but adds a visualization of the training pipeline as a whole.
> >
> > ## Response to Point 1.1
> > Again I request you highlight all changes in blue for full transparency.
> >
> > ## Response to Point 1.2
> > Firstly, let me reiterate that my main two comments in this point was that **(a):** the writing of M2M is not as clear as PPMA (M2M structurally follows the main parts of PPMA, but M2M lacks the same background and context that PPMA provides), and **(b):** Table 1 in M2M is the same as in PPMA but with M2M added. Are there no other baselines that could be added, either that PPMA missed or that have been published since PPMA.
> >
> > The authors' response firstly points me to Weaknesses 1 and 4 vWlp. As an aside, Weakness 1 of vWlp just directs me to part 2 of C3CH, and while I am fine with addressing common reviewer comments with a single response, this convoluted way of jumping around made it very difficult to follow how you are answering my question directly. Regardless, part 2 of C3CH only partially answers part (b) of my question - C3CH gives some references that provide some standard privacy leakage results and the authors respond with an experiment and that they will add it to the main paper. While somewhat valuable, it does not fully answer my question of if Table 1 truly encapsulates every possible (or at least most of the best) baselines for comparison to SMPLy Private. Weakness 4 of vWlp again redirects me to another reviewers response (reviewer 1a4G, not reviewer 1aVg as the authors refer to) and each of the reviewer's questions are not entirely related (reviewer vWlp ask about different architecture/training setups, reviewer 1a4g asks about technical contribution and reliance on off-the-shelf models, I am asking about comparing against more published privacy-preserving action recognition methods (synthetic and/or non-synthetic).
> >
> > I am then pointed to Weakness 1, 2, and 4 of C3CH. Weakness 1 discusses computational complexity which has nothing to do with **(a)** or **(b)**. Weakness 2 was already indirectly referred to in vWlp and is redundant. Weakness 4 is completely irrelevant to both **(a)** and **(b)**. Comparing to other synthetic methods is referenced, but not to the detail I am asking in **(b)**.  On top of all of this confusion, the most important part is that **(a)** is not addressed whatsoever.
> >
> > ## Response to Point 2
> >
> > The author's quote "[...] our work introduces significant innovations that extend beyond a mere replacement of synthetic data with mesh-extracted data. Unlike PPMA, which leverages generic synthetic data, SMPLy Private utilizes SMPL-X meshes, offering a structured and anatomically accurate representation of human motion and posture." directly contradicts itself. I agree that using meshes provides benefits for privacy-preserving action recognition (as I mention in the strengths section), but I am saying that while combining off-the-shelf methods to achieve this is interesting, this alone is not enough to motivate sufficient novelty. I do acknowledge the novelty of K-NEXUS and suggest finding ways to better incorporate similar ideas  in the rest of M2M's pipeline.

---

> > > ### Author Response · Authors · 2024-11-23
> > > **Replies to Points 1, 1.1, and 1.2.**
> > >
> > > # Reply to Point 1 and 1.1 #
> > > Thanks for the suggestion! Moving forward, all revised sections will be highlighted in blue to clearly distinguish them from the original submission. We have now made this change, please see the updated paper.
> > > _____
> > > # Reply to Point 1.2 #
> > > Let us apologize for the confusion. We now understand what you are asking and hope our response is more precise this time. For (a), we now know that we must refine the methodology as it is not as clearly written as PPMA, making it difficult to understand the role of M2M during training. To address this, we have now done the following:
> > >
> > > * The introduction is refined to set the scene of our SMPLy Private framework and M2M augmentation pipeline right before we summarize our contributions / under the hero image (Figure 1).
> > >
> > > * We have added a section (now section 3.1) explicitly comparing M2M and PPMA, noting where M2M improves and explaining how the processes differ in sufficient detail. We have avoided assuming prior knowledge of PPMA.
> > >
> > > * Toward the section of Section 3, in subsection 3.4, we have written a paragraph titled “Putting it All Together” that should connect the dots for the reader who has, at that juncture, almost completed reading the entire section.
> > >
> > > * We have revised the text to explain how M2M ties into Table 1, explicitly noting that M2M is used for both pretraining and alignment, and further hit this point home with a brief note at the start of section 4.
> > >
> > > * Furthermore, Section 4.1 has some additional discussion that should further remind the reader of what we are doing in our approach and again make it clear.
> > >
> > > Hence, we have provided more context and background clarity in SMPLy Private (all changes are in blue, as requested). We have made these parts more self-contained, so readers don’t need to refer back to PPMA. We do vehemently agree that your suggestions on this have further made our paper more straightforward to read / more precise; if more is needed, please let us know specifically more about what you are looking for, and we will address/make the changes ASAP before the end of the rebuttal/discussion period.
> > >
> > > We point to other reviewers partially for part (b) here because there are no additional works (that use synthetic or otherwise) that have followed PPMA, which is SOTA in this domain of privacy preservation. We beat it on all fronts across all datasets using meshes instead of generating/using entirely new data samples with synthetic video-game-like data as we argue features learned from scenes in such synthetic datasets, while useful, cannot capture various nuances from real-world scenarios as seen in Kinetics, and then also further transferring to downstream. This way, we retain scene and object features from the original dataset in our approach. Instead of curating new objects, subjects, and scenes altogether – we superimpose a mesh where the human should be by taking them out. During our literature search, we tried to find comparable works given our objectives of human privacy preservation + SSL pretraining, but, to the best of our knowledge, SynAPT, PPMA, and our work are the most comprehensive. Other works like the one you have pointed to, for instance (i.e., SPAct), only do their evaluations on 2-3 datasets (only one of which overlaps: UCF101). However, after applying their framework, SPAct only achieves ~60% top-1 accuracy on UCF101, which pales compared to PPMA and SMPLy Private (both approx. +25-30% better). Hence, we felt it was appropriate to include more “competitive” models, such as those listed in SynAPT and PPMA. Implementing and re-running other baselines (outside of SPAct, as we did conduct experiments with their framework – we can report those scores if need be in Table 1) across all the datasets we used would be a huge effort in our training scheme for them only to fall far short of both PPMA and SMPLy Private. While we do not capture every possible baseline/method in Table 1, we are comparing ourselves to the current best-published baseline (as you point out), PPMA. Again, we apologize for the miscommunication on our part and hope now this clears up your concern.

---

> > > ### Author Response · Authors · 2024-11-23
> > > **Reply to Point 2**
> > >
> > > # Reply to Point 2 #
> > >
> > > That is a fair assessment. We want to clarify our quote. We were saying that the meshes also help better feature learning as we improve performance over synthetic methods like PPMA while deterring biases related to race and gender (the latter of which we attempted to assess). While we cannot argue sheer novelty, we would like to think that we do have a significant and worthy contribution as an applied combination of methods that beat prior works in many ways – setting a new benchmark and standard for SSL pre-training with privacy preservation, making the shift from synthetic data to “SMPLy” augmented data :). If anything, we refer you back to the updated Section 3.1 of our paper to explain our point on how our framework is, at least, somewhat novel in its approach but perhaps not a novel methodology that has been built from scratch as you and the other reviewers might be expecting/looking for.

---

> > > > ### Comment · Reviewer_8VCT · 2024-12-02
> > > > **Reviewer Reply to Point 2**
> > > >
> > > > Again, I understand that the meshes serve to anonymize private attributes and also improves the learned representations. My original point is that this idea alone (and the propose pipeline) is not a strong enough for acceptance in my opinion. Usually limited novelty is not a sufficient, single reason to reject a paper, as extensive analyses, strong improvements over the baseline, and/or insightful experiments can also serve a benefit to the community. However, as I mentioned in my previous response, I don't believe this paper possesses any of those aspects (see Point 3).

---

> ### Author Response · Authors · 2024-11-20
> **Rebuttal to 8VCT: Part 2/3**
>
> # Addressing Point 3 #
>
> This concern is quite related to what reviewer `vWLp` raised (**please refer to Part 3 of our response there, titled "Addressing Question 7 and Question 8"**). We strongly encourage you to read that alongside this response. Your concern about the distinction between gendered and non-gendered meshes is valid and crucial. These distinctions are outlined in our methodology, where we explain that gendered meshes are instantiated based on gender labels, while gender-neutral meshes lack any specific gender characteristics.
>
> We acknowledge that this explanation could be more explicit and supported with additional examples and definitions. We will address this by including these details in the Appendix in the final version of the paper. These additions will include differences in mesh characteristics, classes that exhibit gender bias, and a manual count we conducted to substantiate this (**we have now included the gender splits in the revised supplementary material for your reference**).
>
> Additionally, we agree that comparisons with privacy-preserving baselines can provide valuable validation. For this reason, we evaluated our work on VISPR attributes (**again, see Part 2 of our response to reviewer `C3CH`). However, we also emphasize that our work uniquely focuses on exploring the potential of meshes for gender bias mitigation—a novel frontier not explicitly addressed by prior methods. Including benchmarks like PPMA or other privacy-preserving approaches that do not address gender bias would dilute the emphasis on this key aspect. To our knowledge, no existing privacy-preserving data augmentation framework addresses gender bias in tandem with privacy considerations.
>
> Finally, the claim that the "stick-out" nature of meshes improves action recognition performance conflates distinct issues even though what you say could be true (and will definitely be a point of analysis to add to our final version of this paper!). The observed improvements in gender bias mitigation stem from the neutralization of demographic cues, not merely enhanced visibility. We demonstrate this by training on data consisting of humans without meshes, where scores reflect these effects (as shown in the table presented in Part 3 of our response to reviewer `1aVG`). The unbiased representation achieved through meshes addresses systemic challenges in demographic-specific tasks, underscoring their value in mitigating gender bias and closing the realism gap.
>
> --------
>
> # Addressing Sub-point 3.1 #
>
> The accelerated representation learning discussed in Section 4.4, while not directly correlating with final performance metrics, provides a novel perspective on the training dynamics of models on anonymized data. This insight highlights the efficiency of using M2M-augmented data in resource-constrained environments where computational efficiency is crucial. Notably, these findings suggest that SMPLy Private-trained models are inherently better suited for early-stage deployment. Although the final epochs achieve real-data baseline performance, the shorter training curve for SMPLy Private models reveals an underexplored advantage, paving the way for further research in low-resource optimization and early deployment strategies.
>
> To enhance the rigor of our approach based on your suggestion, **we extended training by an additional 50 epochs. Under these slightly more resource-intensive conditions, our model surpassed VideoMAE with real data by approximately 1.1%. We appreciate the reviewer’s suggestion, which motivated this extended experiment**. The new training graphs reflecting these results will be included in the final version of the paper. Additionally, we address this point in our response to reviewer `1aVG` (**see the table in Part 3 of that rebuttal**) as it is worth noting that ViT-S using VideoMAE v2 also outperforms our setup when trained on real data. Therefore, our framework demonstrates significant utility in low-resource settings while offering modest gains when more computational resources and training time are available.

---

> ### Author Response · Authors · 2024-11-20
> **Rebuttal to 8VCT: Part 3/3**
>
> # Addressing Sub-point 3.2 #
>
> The critique that comparing M2M to NH Kinetics + Synthetic from PPMA is more apt overlooks the distinct aims of this work. While PPMA’s synthetic data serves as a generic benchmark, the M2M pipeline specifically addresses realism and to some degree demographic bias without relying on synthetic data. Furthermore, the critique on NH Kinetics lacking humans performing actions is addressed inherently in our methodology—our meshes are designed to replicate human actions more authentically, bridging the realism gap that PPMA’s approach does not do entirely. The alignment of our results with the real Kinetics baseline, coupled with bias mitigation, emphasizes the dual contribution of M2M in both performance and ethical considerations. In any case, you are right that we should more distinctly compare M2M with NHK + Synthetic, but that is what is done in Table 1 one anyways (PPMA vs Ours). Let us know if you would like this to be added within the table in Section 4.5 as well.
>
> ----------
>
> *Thank you so much for your valuable feedback and insights, which have significantly contributed to enhancing our paper. We sincerely apologize if our response caused was disorienting, particularly as navigating this page and reviewing overlapping points with other reviewers may be cumbersome and require a lot of effort on your part. However, we do appreciate it a lot! Lastly, we are fully committed to incorporating all necessary revisions in the final camera-ready version of the paper. With these improvements and our thoughtful responses, we kindly ask you to consider reevaluating your score, as we have diligently addressed all key concerns raised. Thank you again for your time and consideration.*

---

> > ### Comment · Reviewer_8VCT · 2024-11-22
> >
> > ## Response to Point 3.2
> >
> > From my understanding, the alignment stage means you take a model pre-trained in a self-supervised manner, and fine-tune it on the same dataset in a supervised manner. Thus, the model learned generally important features from self-supervision and direct action information from supervised training with ground truth labels. My point is that in Table 4, the authors are claiming that M2M Kinetics is better for alignment since it performs better and closes the realism gap, however they are comparing against NH-Kinetics. If there is no human performing the action during alignment, what exactly is the model being aligned with? It is probably using the background or surrounding objects it has erroneously been aligned with to perform action recognition. This is why I suggested using NH-Kinetics+Synthetic in the alignment stage for your baseline in Table 4, as that would better prove that M2M is better for alignment and closing the realism gap while maintaining privacy. Note that this is still not the same as the PPMA row in Table 1.
> >
> > Overall, I think very few to none of my comments were sufficiently addressed. Thus, I retain my score for now as I await the author's response.

---

> > > ### Author Response · Authors · 2024-11-23
> > > **Reply to 3.2**
> > >
> > > # Reply to 3.2 #
> > >
> > > We now understand what you meant, sorry for misinterpreting. We’ve added the baseline where NH-Kinetics is used only for MAE first and Synthetic is only used for Alignment afterward. We hope this point is now fully addressed.
> > >
> > > _____
> > > *Again, we would like to apologize for the back and forth on this due to our misinterpretation of the initial review by you – but we hope now everything is rectified.*

---

> ### Comment · Reviewer_8VCT · 2024-11-22
>
> ## Response to Point 3
> Firstly, I believe the quote:
>
> > "[...] the distinction between gendered and non-gendered meshes is valid and crucial. These distinctions are outlined in our methodology, where we explain that gendered meshes are instantiated based on gender labels, while gender-neutral meshes lack any specific gender characteristics."
>
> is incorrect, as I am not able to find a description of gendered vs. non-gendered meshes in the entire paper, let alone the methodology section. Moreover, I am not able to find the description regarding these meshes from the quote:
>
> > "We will address this by including these details in the Appendix in the final version of the paper. These additions will include differences in mesh characteristics, classes that exhibit gender bias, and a manual count we conducted to substantiate this (we have now included the gender splits in the revised supplementary material for your reference)."
>
> Again I suggest highlighting changes in the updated pdf to see what was absent in the original draft and to find the changes easier.
>
> > "However, we also emphasize that our work uniquely focuses on exploring the potential of meshes for gender bias mitigation—a novel frontier not explicitly addressed by prior methods. Including benchmarks like PPMA or other privacy-preserving approaches that do not address gender bias would dilute the emphasis on this key aspect. To our knowledge, no existing privacy-preserving data augmentation framework addresses gender bias in tandem with privacy considerations."
>
> If the first sentence is true, wouldn't M2M outperform previous methods on gender-biased action recognition when compared against each other? That is exactly what I am suggesting in Point 3.
>
> Regarding the last paragraph, I believe Table 3 in the main paper is measuring action recognition performance on samples where women are performing male-dominated actions and vice-versa. The idea is that a higher action recognition performance indirectly implies gender-bias mitigation, as the model is focusing solely on the action irrespective of who is performing it (i.e., a woman playing football). My point is that the increased accuracy reported for SMPLy Private could have less to do with gender-bias mitigation, but rather that meshes stick-out more than regular humans. I understand that human meshes could play a part in gender-bias mitigation, leading to higher action recognition in tandem with my "stick-out" conjecture, but Table 3 does not sufficiently explore how much each of these points contribute to the reported numbers.
>
> ## Response to Point 3.1
>
> > "This insight highlights the efficiency of using M2M-augmented data in resource-constrained environments where computational efficiency is crucial. Notably, these findings suggest that SMPLy Private-trained models are inherently better suited for early-stage deployment. Although the final epochs achieve real-data baseline performance, the shorter training curve for SMPLy Private models reveals an underexplored advantage, paving the way for further research in low-resource optimization and early deployment strategies."
>
> This is in direct contradiction to your first response to reviewer C3CH.
>
> > "we extended training by an additional 50 epochs. Under these slightly more resource-intensive conditions, our model surpassed VideoMAE with real data by approximately 1.1%. We appreciate the reviewer’s suggestion, which motivated this extended experiment."
>
> Just for clairty's sake, you extend both M2M AND VideoMAE by 50 epcohs? So you train both methods for 250 epochs and that is when you see your method outperform VideoMAE by $1.1\\%$? If so, this still does not address my earlier point that faster learning in the early stages of training is irrelevant if the best-performing epoch is at the end of training, especially considering I do not find the resource-constrained argument sufficiently convincing.

---

> ### Author Response · Authors · 2024-11-23
> **Replies to Point 3 and 3.1**
>
> # Reply to Point 3 #
>
> We apologize for the confusion; we updated the paper with the wrong draft. The revised version should have the mesh descriptions in the newly added Appendix D and the gendered class splits into .txt files within the updated supplementary materials. We hope this is sufficient; again, the changes are in blue.
>
> We now also understand that you would like for us to compare action recognition performance given gendered classes with other methods. We conducted those experiments but chose not to include them in the paper as we wanted to show more of the efficacy of the meshes. Our experiment was more self-serving in that if the appropriate meshes are chosen, the performance improves with a middle-ground/balanced approach with the neutral/gender-agnostic mesh. However, we can compare PPMA and SPAct (the two we looked into for this); on average, their performance was up to ~10% worse. We can add this to the final paper if required.
>
> Furthermore, on the “stick out” point, we are glad you mentioned that again, as we believe it was a worthwhile investigation. The idea during our discussion with you was to change the neutral mesh’s color (we tried beige and a rainbow gradient) and see that beige had a drop in performance by 0.2 from the white mesh we reported scores on (from 83.1 to 82.9), whereas with a rainbow gradient, the performance increased from 83.1 to 83.5. So, there is some credence to your hypothesis on the meshes sticking out, which is directly tied to the type of color of mesh chosen. We can include these findings (and a few more on male and female meshes, too, with various other colors as well) in the final version of the paper. I hope this sufficiently addresses your concern. Again, thank you for pushing us to make this paper more complete and compelling with these fascinating new perspectives!
>
> ________
>
> # Reply to 3.1 #
>
> The additional 50 epochs were done when pre-training on M2M kinetics. The supervised alignment and downstream evaluation epochs were still fixed at 50 and 30, respectively (so we changed the 200 epochs from SSL pre-training to 250 if you look at Table 5 in the Appendix). Hence, if we train longer, the final/best epoch is at the end of training, as you mentioned. Under these conditions, SMPLy Private outperformed VideoMAE by 1.1%. While early-stage efficiency is valuable in resource-constrained scenarios, this result underscores M2M's ability to achieve superior final performance even under extended training budgets. But we are re-iterating ourselves here again a bit.
>
> However, faster early-stage learning does not replace final performance as a critical metric. However, it is helpful for time-sensitive or low-resource settings where competitive performance must be achieved with fewer training epochs. We propose treating early convergence as a secondary advantage that complements M2M's final solid results. But if this is not convincing, we suggest moving this part of the paper into the appendix as an additional exploratory finding and instead replacing it with our results on VISPR, as we now feel that is a more appropriate experiment for the context of this paper (i.e., ensuring low data leakage across various privacy attributes > faster representation learning in early stages < 100 pre-training epochs). We hope this is a good trade-off for improving the paper in your opinion.

---

> ### Comment · Reviewer_8VCT · 2024-12-02
> **Reviewer Reply to Points 1, 1.1, and 1.2.**
>
> Thank you for considering my suggestions. I believe the writing is more clear now that there are specific sections which better explain the problem statement, previous works, and the benefit of your proposed method. I also appreciate the explanation regarding other baselines and agree that it seems SynAPT and PPMA seem to be the only current relevant work. I believe these specific concerns of mine have been addressed.

---

> ### Comment · Reviewer_8VCT · 2024-12-02
> **Reviewer Reply to Point 3 and 3.1**
>
> The authors note:
>
> > "[..]  you would like for us to compare action recognition performance given gendered classes with other methods. We conducted those experiments but chose not to include them in the paper as we wanted to show more of the efficacy of the meshes. Our experiment was more self-serving in that if the appropriate meshes are chosen, the performance improves with a middle-ground/balanced approach with the neutral/gender-agnostic mesh"
>
> Is the paper's main contribution not about privacy preservation? I would think experiments that further exhibit that your proposed method decreases gender or race bias as opposed to previous methods would be integral to the paper. It is great that your experiments with previous methods seemed to not perform as well, but I think further investigation into how your method mitigates gender bias in general should be preferred over investigating which meshes improve action recognition performance. I acknowledge that the authors ran some experiments where your method improved over baselines by $10\\%$ on average, but similar to my feelings on many other points, more detail and time would be needed to incorporate these experiments into the story of the paper (especially since very little detail is given about this $10\\%$ improvement). I believe the experiments addressing the "stick out" hypothesis are great and I appreciate that the authors addressed that, but again it will require a major change to the story/overall work to incorporate that experiment and discuss its impacts.
>
> Regarding Point 3.1, I still feel the same concern is present. If you train for a really long time, you eventually improve over VideoMAE by a marginal amount, which is alright. But the whole section is about early representation learning, and the claim about resource-constrained settings is not supported well enough and still contradicts the discussions with Reviewer C3CH regarding computational complexity of the proposed pipeline. Moving this entire section into the Appendix and replacing it with an entirely new VISPR section that was only considered during the rebuttal period is too major of a change in my opinion - it will be a good addition to the paper but will require a resubmission.

---

> ### Comment · Reviewer_8VCT · 2024-12-02
> **Reviewer Reply to Point 3.2**
>
> This still does not fully address my concern. The point of Section $4.5$ is to claim that M2M Kinetics is a better dataset for alignment than synthetic data. The way the authors initially presented the table was to pretrain all baselines on M2M Kinetics then performing alignment on different datasets. The added row does not address this concern, as I was requesting Stage 1 pre-training on M2M Kinetics and Stage 2 alignment on synthetic to properly evaluate M2M as an alignment dataset.
>
> **In summary, some of my concerns were addressed, some concerns were partially addressed but would require major changes to the paper, and some concerns are still misunderstood/not addressed. I believe through this rebuttal process that the paper has improved in quality, and with the additional feedback from other reviewers, the authors have plenty to add to the paper for a future resubmission. I will retain my score as it is still the highest among other reviewers, and I believe the paper is still truly marginally under the threshold for acceptance.**

---

### Official Review · Reviewer_vWLp · 2024-11-02

**Soundness:** 2
**Presentation:** 2
**Contribution:** 1
**Rating:** 5
**Confidence:** 4

**Summary:**

The authors present a privacy-preserving augmentation framework that replaces human subjects with realistic 3D meshes. The method effectively mitigates bias and privacy concerns related to the human subjects without reducing the utility performance. The paper provides insight to how this replacement affects pretraining and finetuning stages. The authors additional propose a class sampling procedure to guarantee diverse classes for efficient training.

**Strengths:**

1. The motivation and high-level ideas for this work is very strong. It makes a lot of sense that replacing diverse humans with a standardized subject would alleviate many privacy and bias concerns.
2. Results seem fairly strong and back up major claims about utility.
3. The point that using this augmented data can increase learning speed is moderately interesting.

**Weaknesses:**

1. There is no measure of privacy-utility tradeoff in Table 1. While replacing the human entirely is a strong form of privacy-preservation, it is not sufficient to just assume perfect privacy in this scenario. For example, a model may still be able to perceive gender through analyzing motion patterns, not just appearance. It would be helpful to see a quantitative tradeoff that can be used to compare between methods.
2. The K-NEXUS clustering algorithm is comprehensive and potentially interesting, but feels like an unnecessary contribution. I understand why it is effective, but a fair comparison would be using the same subset of classes as previous works. If the goal is not a direct comparison with prior work subsets, then why use a subset to begin with? Computation? Additional details to clarify these points would be useful.
3. One of the major claims is 'faster learning speed' in Section 4.4, which sounds useful, but this may be offset by preprocessing computation time.
4. Only one architecture/training setup (VideoMAE ViT-B) is shown, so it's hard to tell if the same findings would apply to more architecture types/training styles. Additional experiments would better support the paper claims.
5. It seems like the human detection occurs framewise (Section 3.2, Line 211). This may result in some unnatural videos if some frames miss a detection. It may be more natural to choose a video segmentation model to propagate masks instead of relying on many separate detections. The tennis GIF in the supplementary did look pretty good, but it would be more convincing to see video results from more complicated scenes.
6. The overall technical contribution feels weak here. The dataset construction is interesting, but as far as I can tell, there is no unique method proposed that goes with it, just basic MAE pretraining/fine-tuning.
7. On the minor side of things, the ICLR format mandates the table number and caption appearing before the table, not after.

**Questions:**

1. In Line 251, how are the segmentation masks for each object acquired, including the human subject? The text references a method specific to the human subjects, but not objects. Is there a separate segmentation model you use, so you would have two masks for each human? Please clarify this.
2. Can the authors provide some computation/runtime analysis for the preprocessing steps? How long does it realistically take for a single video, maybe like 10 sec/300 frames?
3. For K-NEXUS, since you are already using a LLaVA encoder, why not use the text encoder to encode the class names? In theory, the class videos and text names should have similar representations. It would be interesting to see a comparison between the classes chosen using the visual representations vs. the textual representations.
4. What is the difference between SMPLy Priv and SMPLy Priv w/ K-NEXUS? Is without a disjoint set of 150 random classes? So not just the K-NEXUS classes were annotated, but a separate set of classes as well? Please share more information on these classes, there is no list of selected classes anywhere in the paper or supplementary material.
5. See W5, is there anything done to handle disjoint human detections in the videos? Additional qualitative videos for multi-human scenarios would be helpful.
6. Line 323-324: "with SMPLy Private and the use of our M2M-augmented dataset..." What exactly is SMPLy Private then? I was under the impression that the dataset was the contribution and the model was VideoMAE.
7. How do you define 'male-biased' and 'woman-biased' classes in Table 3? How many of each are chosen/what chooses them? Why not just look at subclass performance across all classes? More details here would help justify this experiment.
8. Following up with Q6, what is the difference between the male, female, and neutral meshes (Table 3)? This is likely described in a previous paper, but it is an important detail to these findings and more detailed explanations of these is necessary for this paper.
9. Line 52: Claim (2) references that the paper explores the potential for mitigating background and scene-object related biases, but I don't see any further explanation for background/scene-object interactions. Is there additional support for this/performance on bias-related benchmarks?

### Final Thoughts
Overall, I truly believe this paper has a lot of potential, but I would not recommend it for acceptance in its current state. The core ideas are strong, but there are a lot of details that need clarifying. A solid technical contribution for better learning with these meshes instead of using basic MAE pretraining would definitely flip my rating for this paper.

---

> ### Author Response · Authors · 2024-11-20
> **Rebuttal to vWLp: Part 1/4**
>
> # Addressing Weakness 1 #
>
> This is the same concern brought up by reviewer `C3CH`. **Please refer to Part 2 of that rebuttal**. We hope this sufficiently answers your concern on this too.
> _____
>
> # Addressing Weakness 2 #
>
> This is the same concern brought up by reviewer `1aVG`. **Please refer to Part 1 of that rebuttal (titled “Addressing Dataset Weakness 1”)**. We also encourage you to read that particular rebuttal as a whole because it will more than likely answer all your questions on K-NEXUS in general.
>
> _______
>
> # Addressing Weakness 3 and Question 2 #
>
> Again, we ask you to kindly look at our response to reviewer `C3CH`. **Please refer to Part 1 of that rebuttal** for the run-time analysis. Furthermore, please kindly see our response to reviewer `8VCT` where we address their **Sub-point 3.1**. We hope these responses put together should sufficiently ensure your questions on this part of our paper are addressed.
>
> _______
>
> # Addressing Weakness 4 #
>
> This is a somewhat similar issue brought up by reviewer `1aVG`. **Please refer to Part 2 and 3 of that rebuttal (titled “Addressing Technical Weakness 1” and the following table with the various MAE-ViT setups)**. Again, we hope this suffices.
>
> _________
>
> # Addressing Weakness 5 #
>
> We acknowledge this limitation, as noted in our conclusion, since we do not utilize video instance segmentation (VIS). However, this issue affects only a small portion of the videos (**refer to Figure 8 in the revised Appendix B.1 and Part 4 of our response to reviewer** `1aVG` **under "Addressing Technical Weakness 2"**). Currently, no VIS methods are compatible with SMPL-X; they only support SMPL. Adopting SMPL, however, would mean sacrificing expressions and limb details, which are crucial not only for preserving privacy but also for action recognition. This trade-off was a necessary decision. Developing a VIS-compatible approach for SMPL-X is a substantial undertaking deserving of a separate paper and was beyond the scope of our work. That said, we are happy to provide more complex examples if requested. Let us know if you’d like us to include additional .gif/.mp4 examples in the supplementary materials for the final version of the paper.
>
> ________
>
> # Addressing Weakness 6 #
>
> Our primary contribution was intended to be the dataset, not a model-centric approach or the proposal of a novel method. Our goal was not to pursue novelty but to adopt an informed approach by synthesizing various literature to address a pressing issue for "social good"—preserving human privacy in video data. This directly ties to our response to Weakness 4 above.
>
> While we have included multiple pretraining methods, we were unable to explore additional encoder architectures due to time and resource constraints. However, we would like to highlight that the VideoMAE [1] paper extensively demonstrated the performance of ViT backbones, showcasing their ability to efficiently and effectively accommodate larger parameter sets. Similarly, this was observed in the Swin Transformer paper [2, 3].
>
> In our final version, we are willing to allocate resources to train a ViT-L backbone as additional work to demonstrate the scalability of our method if the reviewer deems it necessary. However, we **prefer not to**, as we believe the table provided in our rebuttal response to reviewer `1aVG` sufficiently addresses this concern.
>
> We hope the reviewer recognizes that MAE pretraining with ViT backbones is widely accepted as the gold standard for robust video SSL pretraining [1, 4, 5]. Our contribution lies in synthesizing existing methods and developing the M2M augmentation, rather than proposing a new architecture or SSL training regime.
>
> [1] Tong, Z., Song, Y., Wang, J., & Wang, L. (2022). VideoMAE: Masked autoencoders are data-efficient learners for self-supervised video pre-training. Advances in Neural Information Processing Systems, 35, 4093–4104
>
> [2] Liu, Z., Lin, Y., Cao, Y., Hu, H., Wei, Y., Zhang, Z., Lin, S., & Guo, B. (2021). Swin Transformer: Hierarchical Vision Transformer using Shifted Windows. arXiv.
>
> [3] Liu, Z., Hu, H., Lin, Y., Yao, Z., Xie, Z., Wei, Y., Ning, J., Cao, Y., Zhang, Z., Dong, L., Wei, F., & Guo, B. (2022). Swin Transformer V2: Scaling Up Capacity and Resolution. arXiv.
>
> [4] Wang, L., Huang, B., Zhao, Z., Tong, Z., He, Y., Wang, Y., Wang, Y., & Qiao, Y. (2023). VideoMAE V2: Scaling video masked autoencoders with dual masking. Proceedings of the IEEE/CVF Conference on Computer Vision and Pattern Recognition, 14549–14560
>
> [5] Feichtenhofer, C., Fan, H., Li, Y., & He, K. (2022). Masked autoencoders as spatiotemporal learners. arXiv.
>
> _____
>
> # Addressing Weakness 7 #
>
> Thanks for pointing this out! We have made the change, see revised paper.

---

> ### Author Response · Authors · 2024-11-20
> **Rebuttal to vWLp: Part 2/4**
>
> # Addressing Question 1 #
>
> This was an oversight on our part and we sincerely apologize. A previous iteration of our work used a different configuration. Since then, our occlusion aware meshing is complemented by using the Segment Anything Model (SAM) [5]. So yes, it is one model that masks various objects including humans to avoid various occlusions. **This has been updated in our paper**.
>
> [5] Kirillov, A., Mintun, E., Ravi, N., Mao, H., Rolland, C., Gustafson, L., Xiao, T., Whitehead, S., Berg, A. C., Lo, W.-Y., Dollár, P., & Girshick, R. (2023). Segment Anything. arXiv.
>
> _______
>
> # Addressing Question 3 #
>
> Please refer to the updated Section 3.1. We encode both labels (as text) and image frames, and we apologize for any earlier confusion. Using only the text encoder would not be feasible for K-NEXUS, as it would lack the necessary functionality. Clustering identical class labels or text and expecting K-NEXUS to select 150 classes from such a homogenous clustered space would be ineffective. This approach would lack diversity; for example, repeated labels across all classes would lead to redundant clusters (multiplying the number of text labels per class by the number of samples in that class).
>
> To address this, we use image-text pairs to create a shared representation by projecting and aligning both features into the same space. This approach ensures greater diversity, enabling K-NEXUS to function effectively. The image frames are crucial as they introduce differences visually, which then translate into semantic distinctions. We apologize for any lack of clarity in the original explanation and have corrected this in the **updated paper**, addressing any typographical issues as well.
>
> ________
>
> # Addressing Question 4 #
>
> This is a similar concern brought up by reviewer `1aVG`. **Please see Parts 1, 2, and 5 of that rebuttal for a full understanding on K-NEXUS. Furthermore, we have now included the K-NEXUS splits in the supplementary material.**
>
> _________
>
> # Addressing Question 5 #
>
> Please see Figure 1 (e.g., “clean and jerk”, “zumba”) and Figure 2 for multi-human examples. If you require additional examples, we would be happy to include them in the supplementary material for the final/revised version of the paper.
>
> _________
>
> # Addressing Question 6 #
>
> SMPLy Private is what we call our entire end-to-end pipeline (segmentation + mesh recovery + inpainting + VideoMAE pre-training + alignment + downstream evaluation), and M2M is the actual data augmentation method that creates the meshed dataset (i.e., M2M Kinetics). **We have now made this distinction in the updated version of the paper now for more clarity right before Section 4.1**.

---

> ### Author Response · Authors · 2024-11-20
> **Rebuttal to vWLp: Part 3/4**
>
> # Addressing Question 7 and Question 8 #
>
> The algorithms built on SMPL-X utilize all three mesh types (male, female, and neutral) – you can find more details on the differences between these meshes at the SMPL-X paper reference below [6]. Our mesh rendering strategies are designed to assign the appropriate mesh type for a more accurate representation: male or female meshes are used when the person's features in the video are descriptive enough, while a neutral mesh is applied when insufficient information is available. In analyzing action classification datasets, we observed that male meshes were activated more frequently for certain classes, while female meshes predominated in others. This led us to conduct this study to highlight that our models effectively capture dynamics across both genders. Our mesh approach preserves vital gender information without compromising performance, regardless of the mesh type.
>
> The definitions of "male-biased" and "female-biased" classes in Table 3 are based prior analyses of action recognition datasets. Certain actions, such as weightlifting or football, are statistically more likely to feature male participants, whereas others, like ballet or yoga, tend to feature women more prominently. However, these categorizations were informed by dataset annotations during our manual qualitative review (**see Figure 8**). To ensure balanced evaluation, we selected an approximately equal number of male- and female-biased classes (34 to 36), reviewing dataset metadata and consulting prior research. A list of these classes are now included in a supplementary document for transparency.
>
> While analyzing subclass performance across all classes could provide broader insights, this experiment focuses on assessing whether gender-neutral meshes effectively mitigate biases specifically in male- and female-biased classes. This targeted approach isolates the impact of demographic features on model performance, ensuring clearer insights into bias mitigation. Including all classes might dilute these findings, as biases are most evident within specific subsets of data.
>
> Our evaluation on VISPR further supports this analysis (refer to our discussion with review `C3CH`, specifically Part 2). We’ve updated the supplementary materials in which you can now find the gender splits for this experiment. We appreciate this feedback and will extend the discussion accordingly in the revised version of our paper.
>
> [6] Pavlakos, G., Choutas, V., Ghorbani, N., Bolkart, T., Osman, A. A. A., Tzionas, D., & Black, M. J. (2019). Expressive Body Capture: 3D Hands, Face, and Body from a Single Image. arXiv.
>
> ______
>
> # Addressing Question 9 #
>
> The reference to mitigating background and scene-object biases stems from the fact that SMPLy Private deliberately replaces real human appearances with SMPL-X meshes, which inherently decontextualize human actions from specific environmental or object-related cues (which is also further done by K-NEXUS, please see our discussion with review `1aVG` on this. Specifically Part's 1, 2, and 5). This substitution reduces the direct association between human actions and surrounding scene features. For example, in standard datasets, certain actions might disproportionately co-occur with specific object types (e.g., "riding" often appearing with bicycles or horses), potentially biasing models to associate the action with the object rather than the human dynamics. By using human-only meshes devoid of rich texture or context, our method pushes the model to focus on action dynamics rather than environmental correlations. If you refer to the SynAPT [7] and PPMA [8] papers that discuss this, just by outperforming their methods, we effectively are doing a great job mitigating scene-object related bias, especially when K-NEXUS is used. If you require we have this discussion in the revised/final version of the paper, we would be more than happy to include it! Thank you for bringing this up.
>
> [7] Zhong, H., Mishra, S., Kim, D., Jin, S., Panda, R., Kuehne, H., Karlinsky, L., Saligrama, V., Oliva, A., & Feris, R. (2024). Learning human action recognition representations without real humans. Proceedings of the 37th International Conference on Neural Information Processing Systems (NIPS '23), 2839.
>
> [8] Kim, Y., Mishra, S., Jin, S., Panda, R., Kuehne, H., Karlinsky, L., Saligrama, V., Saenko, K., Oliva, A., & Feris, R. (2024). How transferable are video representations based on synthetic data? Proceedings of the 36th International Conference on Neural Information Processing Systems (NIPS '22), 2588.

---

> ### Author Response · Authors · 2024-11-20
> **Rebuttal to vWLp: Part 4/4**
>
> *We sincerely apologize if our response caused any inconvenience, as we understand it may be somewhat cumbersome to navigate this page and review our responses to other reviewers due to overlapping points. Once again, we would like to emphasize that devising a novel pretraining or SSL method was not our primary objective whatsoever. For this reason, we believe MAE-type training for videos is sufficient (the standard pre-training method for video data), and we hope the table of SSL MAE-encoder ablation included in our response to reviewer `1aVG` addresses your concerns somewhat satisfactorily. We kindly ask, if it is not too much trouble, that you reconsider your evaluation of our work in light of the updates and responses provided to both your concerns and those of other reviewers. We are deeply grateful for your time and thoughtful feedback, which have significantly contributed to strengthening our paper. Thank you for your consideration.*
>
> **EDIT / P.S.** -- Here are some examples of contrastive SSL video methods that pale in comparison to MAE-type methods [a, b]. Also, refer to the Kinetics-400 leaderboard and see that most SSL methods are populated by MAE / VideoMAE type pretraining schemes [c].
>
> [a] Qian, R., Meng, T., Gong, B., Yang, M.-H., Wang, H., Belongie, S., & Cui, Y. (2021). Spatiotemporal Contrastive Video Representation Learning. arXiv.
>
> [b] Wang, J., Bertasius, G., Tran, D., & Torresani, L. (2022). Long-Short Temporal Contrastive Learning of Video Transformers. arXiv.
>
> [c] Papers with Code Leaderboard on Kinetics-400 is populated with MAE ViT-based SSL-encoder frameworks: https://paperswithcode.com/sota/action-classification-on-kinetics-400

---

> ### Comment · Reviewer_vWLp · 2024-11-23
> **Rebuttal Response (Part 1/2)**
>
> ## Addressing Weakness 1:
> It is helpful that you were able to run experiments with the VISPR1 protocol. However, I have some concerns about the evaluation. What was your method? Did you replace humans in the images with meshes, then train a classifier on these images? I am very surprised at these numbers, it is a drastic decrease compared to previous results. My major concern is that using an untrained, randomly initialized classifier achieves $\approx$ 25% on VISPR1 due to its binary classification paradigm (label weights aren't balanced, accounting for <50% expected value). I would like some clear rationale for how your method is "fooling" the classifier, causing it to predict the incorrect attributes. The goal of this dataset is to reduce predictions to random chance (~25%), not specificially to cause the classifier to choose the wrong prediction. I believe this would imply that the classifier is able to learn the correct attributes, but choose the wrong one. This may not be 100% true, but the authors need to provide clear rationale for how these numbers were achieved.
>
> ## Addressing Weakness 2:
> Thank you for the response, I have a better idea of the contribution intention now. However, it still unclear to me whether your performance improvements stem from the selected subset or from your M2M augmentation. It would provide more clarity to see a comparison using a subset from a previous work, but with your augmentation. It still seems possible that a previous method using your K150 subset would outperform that of your M2M method. More insight into the differences between the selected subsets would be helpful as well.
>
> ## Addressing Weakness 3:
> Unfortunately, these comments to other reviewer concerns do not address my concerns. In one, you state that "the pipeline is designed for preprocessing datasets on robust computational systems, which are more than capable of handling the computational load", while in the other, you state that "(t)his insight highlights the efficiency of using M2M-augmented data in resource-constrained environments where computational efficiency is crucial". These directly contradict each other, and I am left unable to see the benefit of the faster training from this perspective. Nonetheless, it is still an interesting insight to the learning process with your synthetic data, so it is good to include, but as is, it does not seem to be a valuable benefit.
>
> ## Addressing Weakness 4:
> It is great to see further experiments with different backbones and model sizes, that is helpful. As a friendly note, even if MAE pretraining is the standard and achieves the best performance, it would still strengthen the rigor of this analysis to consider alternate forms of models and pretraining (contrastive for example), but the further analysis did address this concern by expanding upon the model choices.
>
> ## Addressing Weakness 5:
> I apologize if this was not clear, but my concern was more about how missing a detection within a video segment. I see your analysis of specific failure cases, this is good to show, but I am curious as to how this affects the final video. I would expect some unnatural, jittery movement in this case. It may not happen often/not be a major detriment, but I would like the authors to address this possibility and its potential implications.
>
> ## Addressing Weakness 6:
> The response does not better emphasize the technical contribution. I get that the point is to address privacy-preserving action recognition, but my comment is about the technical contribution. The proposed contributions appear disjoint, not building off each other in a real sense, other than just both slightly improving performance from different perspectives.
>
>
> ## Addressing Question 1:
> Makes sense, thank you for clarifying.
>
> ## Addressing Question 3:
> I understand the limitations of only using the text encoder, I just think it would achieve a similar split with much less effort. It would strengthen your contribution if you can show improved performance using your visual information K-NEXUS over the splits given by just encoding the text labels. This may just be a pedantic point, as it makes intuitive sense that your method would be better, so I can let this point rest. However, I am very concerned with your response to this. My impression when first reading this section is that you only used the visual features. There wasn't a problem with this, though using a combination of visual and text seems fine too. My concern is that the updated section now clearly indicates that both text and visual features are utilized. While I do not have the previous version of the paper saved and may have just misread, this change is a major difference. Can the authors please clarify what the original method was, and if the core method is changed, proper analysis of the difference between the two version should be provided.

---

> ### Comment · Reviewer_vWLp · 2024-11-23
> **Rebuttal Response (Part 2/2)**
>
> ## Addressing Question 4:
> It is helpful to see the K-NEXUS class list, thank you for including those. But I am also interested in the other split you used without K-NEXUS. Is this a split from a previous paper? Is it random? It is specifically chosen to be redundant? Is it just the first 150 classes?
>
> ## Addressing Question 5:
> These are just single frame examples, seeing actual videos (where there may be some missed detections/occlusions) is what I was interested in.
>
> ## Addressing Question 6:
> Thank you for addressing this.
>
> ## Addressing Question 7 & 8:
> I understand what a male and female biased class could mean, but I would like to know how these classes were chosen, and your response is vague. Figure 8 does not show manual qualitative review, and I still do not see any justification other than "manual qualitative review". I think this suggests looking at the gendered meshes chosen by the method, but this isn't exactly made clear. Also, it is still not clear what the implications of this gender study is. The results look like best performance is achieved simply by using the appropriate gendered mesh, even in these biased class scenarios. You then state that "this indicates that 3D meshes help mitigate gender-action bias by offering a gender-agnostic representation", but nowhere do you indicate that a gender-neutral mesh is chosen for your work. If this is the better option for gender bias, then why do you not choose to use them? On top of this, I find it very concerning that you state: "Our mesh rendering strategies are designed to assign the appropriate mesh type for a more accurate representation: male or female meshes are used when the person's features in the video are descriptive enough, while a neutral mesh is applied when insufficient information is available." This defeats a major component of privacy-preservation by explicitly guessing subject gender and expressing this in your mesh representation. This also loops me back to my VISPR question. One of the attributes in VISPR1 is gender, so if you explicitly keep gendered informantion, then how is it that you are able to achieve such low performance? Overall, this aspect needs more clarification and justification for revealing subject gender in "privacy-preserving" action recognition.
>
> ## Addressing Question 9:
> This does not address my concern. Your point in the following quote is important and potentially correct: "our method pushes the model to focus on action dynamics rather than environmental correlations", but I believe it needs more analysis and justification than simply 'we outperform previous methods therefore it is true'. I do agree with this statement: "(f)or example, in standard datasets, certain actions might disproportionately co-occur with specific object types (e.g., "riding" often appearing with bicycles or horses), potentially biasing models to associate the action with the object rather than the human dynamics." But I fail to see how this does not apply to your method as well. Even though the meshes "stick out", the same actions still disproportionately co-occur with specific object types and are therefore susceptible to the same biases. An explicit justification to your claims about reducing background and scene-object biases is still necessary.
>
>
> ## Overall
> It is great to see the amount of effort the authors have put in to this rebuttal, but unfortunately, the responses seem to dodge many of the core concerns and make bold claims without proper justification. Therefore, I choose to maintain my rating for now.

---

> > ### Author Response · Authors · 2024-11-24
> > **Reply to Question 9**
> >
> > # Reply to Question 9 #
> >
> > While specific ablations have not been done on this point, based on prior works like SynAPT and PPMA, we know that there is a higher scene-object bias from the left-side of our Table 1, starting at UCF101, which reduces to lower object-scene bias datasets like UAV-Human (see Figure one in the SynAPT paper: “How Transferable are Video Representations Based on Synthetic Data?”). We are going based on this. If this is not substantial enough evidence, we acknowledge that and relent this point, but we would appreciate your understanding nonetheless (note, we do well on higher scene-object datasets downstream, but struggle a little with UAV-Human relative to PPMA showcasing that SMPLy Private is indeed better for higher scene-object bias tasks).
> >
> > ____
> >
> > *We again thank you for your invaluable feedback and hope to have clarified your concerns well enough for a potential reconsideration of our paper's current score. We know how time-consuming this process can be so we've tried to be proactive in our responses while appreciating your time. Thanks again!*

---

> ### Author Response · Authors · 2024-11-24
> **Reply to Weakness 1, 2, and 3 + Question 4**
>
> # Reply to Weakness 1 #
> Very embarrassing on our part and we sincerely apologize!! There happened to be an implementation issue that outputted the final result inappropriately by a factor of 3, so the cMAP scores without and with K-NEXUS are actually 33.9 and 38.1 respectively (we have updated this in the other response to reviewer `C3CH` as well). Note, in the latter case we just use the gendered-mesh which is most occurring after K-NEXUS is applied on Kinetics (e.g., if that is the neutral mesh, that is what is applied on VISPR1). Yes, our approach involves replacing humans in the images with SMPL meshes using our M2M framework. The meshes are then superimposed onto the inpainted backgrounds to create anonymized versions of the VISPR1 dataset. We trained a classifier on the anonymized images to evaluate privacy leakage using the VISPR1 protocol. The reported cMAP scores reflect the classifier's ability to predict attributes from these anonymized images. Our reported score indicates that while the classifier is not entirely "fooled," it performs only slightly above chance on the anonymized dataset. This suggests that the SMPLy Private method substantially obfuscates the key attributes used for attribute inference while not perfectly reducing the signal to random noise.
>
> We also offer some rationale for our results:
>
> * Despite anonymization, SMPL meshes inherently encode body proportions and approximate pose, which may inadvertently correlate with some privacy-sensitive attributes (e.g., height could correlate with gender in some contexts).
>
> * We posit the inpainting process retains the original scene, which might allow the classifier to use non-human features as weak proxies for certain attributes in VISPR1 which are all human-based. For example, scene objects or locations in the dataset may correlate with specific demographics.
>
> * A score above chance (but below previous benchmarks) could indicate that the classifier is capturing residual patterns but cannot robustly infer the true attributes. This is supported by the fact that the meshes are designed to be neutral, removing direct cues like skin tone and particular facial features. However, the classifier may rely on remaining weak signals or mislearn spurious correlations, resulting in slightly elevated cMAP scores.
>
> In general, while we know the goal of privacy-preserving methods is to reduce attribute inference to random chance (25%), achieving this requires a more thorough elimination of all residual cues, including indirect ones from background and pose. Our results show that the SMPLy Private method significantly reduces attribute predictability compared to unmodified data and prior benchmarks, but we acknowledge that additional steps, such as further contextual obfuscation and/or enhancing mesh standardization, could further align the scores with the ideal random baseline.
> _______
> # Reply to Weakness 2 and Question 4 #
> In Table 1, we make this distinction now very clear. SMPLy-Private uses the K-150 sub-set that was curated by the work we built upon (PPMA and SynAPT), where we have noted in this discussion phase that they randomly selected the 150 classes (we now mention this at the start of Section 3.2). Everything in Table 1 uses that set-up of 150 classes excluding the last row where SMPLy Private uses the classes selected by K-NEXUS (currently included in the supplementary zip file). Our boosted performance with K-NEXUS (“SMPLy Priv. w/ K-NEXUS”) essentially builds on-top of our boosted performance with the M2M mesh augmentations (“SMPLy Priv.” from Table 1 which uses the K-150 sub-set from PPMA/SynAPT). We hope this clarifies your concern entirely now.
>
> This should also answer Question 4: We hope that we have addressed this now -- essentially, it is just the splits used from the SynAPT/PPMA papers.
> _____
> # Reply to Weakness 3 #
> While you can refer to our new response to reviewer `8VCT` titled “Reply to 3.1”, we have come to the conclusion it might be best to shift this experiment (since it is more exploratory) into the Appendix as an additional finding, and shift the VISPR results into the main paper here instead as we believe that you are right in that we should include such results. It is more important, within the context of our paper, to discuss potential data privacy leakage over faster learning in a lower resource setting. Let us know if you are happy with this, thanks!

---

> ### Author Response · Authors · 2024-11-24
> **Reply to Weakness 5 and 6 + Questions 3, 5, 7, and 8**
>
> # Reply to Weakness 5 and Question 5 #
>
> Thank you for clarifying your concern regarding the potential impact of missed detections on the final video. You are correct that missed detections during human removal may occasionally lead to unnatural or jittery movements, particularly when residual artifacts persist across frames. We have observed that such issues are rare due to the robustness of the inpainting and segmentation steps, but when they do occur, they may slightly disrupt temporal smoothness in specific regions of the video. While these artifacts are unlikely to affect downstream performance significantly (as the model learns to generalize across noise during pretraining), we acknowledge that they could be more noticeable in fine-grained, motion-sensitive tasks. To address this, we will include an explicit discussion of this potential limitation in the camera-ready paper, along with examples in the appendix to illustrate how such artifacts manifest and their implications for model performance. We appreciate your insightful feedback, which helps us refine both our limitations section and presentation of the paper. We hope this is sufficiently addressing this particular concern of yours.
>
> Furthermore, we have updated the supplementary materials with an example (see under the folder “failed-example”), this also backs up the validity of your concern in Weakness 5. Again, we will address this in our limitations section of the for the final camera-ready version of the paper.
>
> ____
>
> # Reply to Weakness 6 #
>
> You are correct in pointing out that the more innovative aspect of our paper lies in K-NEXUS, while the other elements represent a creative combination of existing methods. The goal of our paper was not to introduce a novel SSL method (or any other type of novel method) entirely distinct from MAE and using mesh-augmentation for privacy preservation on video data. However, if this is considered a weakness, we respect and acknowledge your perspective. It could indeed be intriguing to explore or develop a method that surpasses existing off-the-shelf solutions in learning mesh representations (something that is uniquely suited to learning mesh forms). This is a valuable suggestion, and we could certainly consider addressing it in a "Future Work" subsection within our conclusion. Would that be sufficient enough to address this concern? Thank you for your feedback on this.
>
> _____
>
> # Reply to Question 3 #
>
> We really appreciate and thank you for acknowledging that this point can rest, and when we did solely use the text encodings in our earlier experiments, the splits were quite random and not consistent. Hence we chose to disregard it altogether. Again, no worries, if need be, we can make this point briefly in the final camera-ready version of the paper.
>
> As mentioned in our response, it was something we needed to clarify further. In our original paper, we assumed that the reader might interpret “label” as the “text label” opposed to the numerical one. This is the only change we made so that it is clear that in our original and only approach on this within the K-NEXUS algorithm was to encode the image-text pairs in tandem. We did not change our approach in this rebuttal period on K-NEXUS. We hope this clarifies, thank you.
>
> _____
>
> # Reply to Question 7 and Question 8 #
>
> Again, we apologize for the lack of clarity in our previous response. Yes you are right, we did choose it based on the method as the SMPL-X meshes do adapt to the human based on identifying their gender. Once we saw this, we looked at the flipped and neutral cases too just by changing the gender parameter of the mesh. To make this process transparent, appended the documentation of the criteria used during this manual review to the revised paper to an additional Appendix D. The implications of this study was to show that we can maintain best performance using neutral meshes and that is actually what we take forward in Table 1 as well. We have now added this in the revised version of the paper (also see footnote 4).
>
> In the quote you bring up, we were just clarifying that the SMPL-X fitting just works that way. However, for VISPR1 we show that after K-NEXUS used on Kinetics, we then apply mesh fitting. The most commonly used mesh superimposed by default is the neutral mesh on K-150, so this is the mesh we take forward for the VISPR1 evaluation. Without K-NEXUS it was the male mesh that most commonly occurred and when applied to VISPR1 that is why the cMAP scores using SMPLy Private without K-NEXUS was approx. 5% higher. Hopefully this clarifies everything now and addresses this concern appropriately.

---

> ### Comment · Reviewer_vWLp · 2024-11-26
>
> **Reply to Weakness 1:** Thank you for the clarification here. These numbers seem much more reasonable and better than existing methods. I would not expect a perfect score here, and your explanation is reasonable.
>
> **Reply to Weakness 2 and Question 4** That makes the most sense. Thank you for clarifying. As reviewer `1a4G` has suggested, a comparison between the K-NEXUS classes and the previously proposed classes would help strengthen your argument.
>
> **Reply to Weakness 3** I do think this change makes sense, given this is poised as a privacy-preserving action recognition work.
>
> **Reply to Weakness 5 and Question 5** Thank you for including this, it is an important limitation, and could be a call to improve video-based detection/segmentation models. It would be interesting to see how the SAM2 [1] segmentation model performs in maintaining the detections, though not crucial to your work.
>
> **Reply to Weakness 6** This was just a single suggestion for a potential contribution that makes sense to me, I do not want you to include it just to address this concern. The main point is that the technical contribution is a bit weak as is, noted by the other reviewers as well. More comprehensive evaluation related to K-NEXUS design choices, outputs, and justification of claims about the benefits of selecting the subset (dataset W.1, reviewer `1a4G`) would help.
>
> **Reply to Question 3** I apologize for this comment then, it appears I had misread it the first time through due to it just saying "LLaVA image encoder". Thank you for clarifying this, the text is a bit more clear now. Since both the image and text are embedded, how are these embeddings combined? Added? Concatenated?
>
> **Reply to Question 7 and 8** The choice of using neutral meshes is most appropriate, thanks for making this clarification. However, then a fair comparison with the previous K150 split and yours using neutral meshes only is warranted.
>
> > "The most commonly used mesh superimposed by default is the neutral mesh on K-150, so this is the mesh we take forward for the VISPR1 evaluation. Without K-NEXUS it was the male mesh that most commonly occurred and when applied to VISPR1 that is why the cMAP scores using SMPLy Private without K-NEXUS was approx. 5% higher."
>
> This claim is not well justified. This just means that the classes chosen during K-NEXUS contains less visually identifiable humans, resulting in neutral meshes. It does not make intuitive sense that the classes chosen by K-NEXUS happens to activate neutral meshes more commonly than without. Regardless, this also does not make sense for VISPR evaluation, which does not depend at all on classes chosen for video pretraining. It is an independent image-based evaluation that should only depend on the choice of mesh. There is no point in comparing Kinetics pretraining sets on VISPR. It should be labelled by the choice of mesh, as the mesh is what is being evaluated. This experiment was asked to evaluate the privacy-preservation capabilities of your model, not your subset selection algorithm.
>
> **Reply to Question 9** I agree that your method _may_ reduce this scene/object related biases, I just don't find the current rationale convincing. Nonetheless, this is not crucial to address, and your provided justifiation is reasonable. I do find your experiment with the color of the mesh (from reply to point 3, reviewer `8VCT`) interesting, this is good to include in the paper. Conclusions drawn from this would provide additional insight into replacing humans with 3D meshes, better supporting your contributions.

---

> > ### Author Response · Authors · 2024-11-26
> >
> > **Response to W2, W6, and Q4.** Of course! Glad we could clarify. On comparing the classes we selected vs. the prior works (PPMA and SynAPT), we show that by selecting classes using K-NEXUS our method further outperforms PPMA and SynAPT in Table 1 (see our latest response to `1aVG` for more info on the matter). Shouldn’t this be sufficient? Unless you want us to list out our selected classes vs. theirs and do a sort of qualitative analysis showing that their classes might have some redundancies that ours don’t see, we used K-NEXUS? We could add more discussion on the design choices, outputs, justifications, etc. for K-NEXUS in camera-ready – we hope now it is very clear what the purpose of K-NEXUS is. However, we do not want it to be that central to the paper to take the focus away from privacy-preservation for SSL pretraining – K-NEXUS was just an additional means to an end. We plan on having a full-fledged paper for the algorithm in the near future at some point.
> > ______
> > **Response to Q3.** No worries, honestly it was on us for having it written like that so we appreciate you helping us clarify it! And yes, embeddings are essentially concatenated.
> > _______
> > **Response to Q7 + Q8.** Of course, we agree with you, this was just an additional step we wanted to take to see if there would be a rationale for choosing what type of mesh to employ during VISPR evaluation. But in general we have found that the neutral mesh gives the lowest cMAP score which we have already reported. In the camera-ready we will simply show this (and how it improves over gendered meshes) vs. other reported measures in the literature like SPAct. We hope this puts your concern to rest, thank you.
> > ________
> > **Response to Q9.** Fair enough – we are glad you find the provided justification reasonable. We are grateful for you looking at our response to reviewer `8VCT`. Thanks!

---

> ### Comment · Reviewer_vWLp · 2024-12-02
>
> **K-NEXUS:** Yes, even more technical/qualitative insight would be useful here. I get that your selected subset achieves better performance, but looking into it, the previous subsets were randomly selected. It is no surprise that your method is better than random. In order for this selection to be a valuable contribution, I believe more insight is necessary. Figure 4 in the appendix is a solid start, but maybe a full embedding plot of all 150 classes would be more convincing than comparisons between two random classes. I still believe an ablation on this selection would better support claims here (visual vs. text vs. text + visual). Also, the original motivation for selecting a Kinetics subset in SynAPT was to balance it out with the size of their pretraining dataset for a more fair comparison. I feel that this class selection makes for an unfair comparison with prior methods.
>
> **Meshes:** I do see that your mesh method does still outperform prior methods on the same subset, but the performance difference is not very strong. More clear benefits of utilizing the meshes over prior methods of privacy preservation, if not a direct performance increase, would better support this method. I recognize that there are attempts at this, but right now the benefits seem minor, computation may be the deciding factor between this and prior work.
>
> **Overall:** This problem is motivating and the solution makes intuitive sense. However, the contributions of this work appear "matter-of-fact" and lack robust justification/exploration. The technical novelty is slim without comprehensive analysis of each component. While many of my concerns have been addressed, not everything from myself and other reviewers have been. It is wonderful to see the amount of effort put in by these authors during the rebuttal and the improvement of the work itself over the course of the rebuttal period, but I still believe that more work needs to be done to convince me that this paper is ready for acceptance. I recommend that the authors spend more time writing a concrete, coherent story to better motivate the contributions, differentiating this work from prior work, and thoroughly analyzing each component of the contributions in the context of a coherent story. I am updating my score to a 5 to reflect my current understanding of the improved work.

---

### Official Review · Reviewer_C3CH · 2024-11-04

**Soundness:** 2
**Presentation:** 3
**Contribution:** 2
**Rating:** 3
**Confidence:** 5

**Summary:**

The paper presents a method attempts at mitigating private attribute information in video by converting humans into 3D meshes. This approach utilizes various off-the-shelf models, starting with human segmentation, followed by replacing segmented humans with 3D meshes and inpainting to remove the original RGB humans. Additionally, it incorporates object information by leveraging another segmentation model.

**Strengths:**

- The paper features clear writing and well-illustrated figures, making it easy to follow.

- The diversity and quantity of action recognition datasets used are commendable.

**Weaknesses:**

- **W1**: A primary concern is that the method contradicts the goal of privacy preservation by anonymizing video with minimal computation, which is essential for deployment on edge devices before transmitting anonymized content to the cloud computation or storage. The anonymization should require less computation than the utility model (e.g., VideoMAE used here). However, the proposed method employs multiple off-the-shelf models- video inpainting, object detection, and SMPL-X, and their combined computational load significantly exceeds that of the utility model. This compromises privacy by exposing private attributes to these models and makes edge computing infeasible.

- **W2**: Another key concern is the lack of evaluation for privacy protection to quantify privacy leakage. In the main comparison Table 1, the method appears to assume that it inherently resolves privacy issues. While 3D meshes might seem to avoid privacy risks at the human perception level, the essential measure is computer perception, which all prior work evaluates [a,b,c,d]. The method should follow standard privacy evaluation protocols, such as training a classifier to detect private attributes from the 3D-mesh-transformed VISPR image dataset, thus quantifying privacy leakage. The results would then indicate whether or not the private attributes are effectively anonymized.

- **W3**: Another limitation is that the approach is restricted to human-related private attributes only. Prior works like [b,c,d] address a broader range of personal identifiable information, including scenes and objects. While this is not a central evaluation point, the authors should acknowledge this as a limitation.

- **W4**: The results on the Diving48 dataset are notably low- 66%, significantly below comparable baselines like TimeSformer-L (81%) and the state of the art at approximately 91%. I strongly recommend that the authors address this low baseline, as the current claims may be misleading. Diving48, as a fine-grained action dataset, includes fast-moving objects where even minor errors in input modalities substantially affect classification performance. It is well-established that 3D meshes are not ideal for representing fast-moving objects, and this is a fundamental limitation of the approach, given its reliance on individual off-the-shelf components that can propagate errors.

[a] "Privacy-preserving deep action recognition: An adversarial learning framework and a new dataset." IEEE Transactions on Pattern Analysis and Machine Intelligence 44.4 (2020): 2126-2139.

[b] "Spact: Self-supervised privacy preservation for action recognition." Proceedings of the IEEE/CVF Conference on Computer Vision and Pattern Recognition. 2022.

[c] "Ted-spad: Temporal distinctiveness for self-supervised privacy-preservation for video anomaly detection." Proceedings of the IEEE/CVF International Conference on Computer Vision. 2023.

[d] "STPrivacy: Spatio-temporal privacy-preserving action recognition." Proceedings of the IEEE/CVF International Conference on Computer Vision. 2023.

**Questions:**

While the method introduces a novel approach to privacy preservation, it has fundamental flaws in its formulation. Specifically, it undermines the objective of efficient and feasible anonymization relative to the utility branch. Furthermore, the absence of quantitative evidence on privacy leakage reduction is concerning. In its current form, I am inclined to recommend against accepting the paper. To address these issues, it would be beneficial for the authors to respond to the following weaknesses (see weakness section for more details):

W1: Provide FLOPs for both the anonymization process and VideoMAE, and compare with prior work [a-d].

W2: Adhere to the privacy protocols established in prior work using the VISPR dataset, and include a detailed analysis. Additionally, provide experimental implementation details.

W3: (Optional) Report results on non-human privacy attributes as well, as done in [b, d].

W4: Improve the baseline on Diving48 and base conclusions on the revised results.

---

> ### Author Response · Authors · 2024-11-19
> **Rebuttal to C3CH: Part 1/4**
>
> # Addressing Weakness 1 #
>
> The primary goal of our M2M pipeline is to provide a method for anonymizing human-based datasets through the use of mesh representations. These operations are not intended to be real-time nor optimized for edge-device deployment. Instead, the pipeline is designed for preprocessing datasets on robust computational systems, which are more than capable of handling the computational load (see Table 5). These operations run locally, ensuring that private content does not leave secure environments. The goal is, that any derived data (e.g., meshes) are stripped of sensitive identifiers prior to external usage. The anonymization process is thus a preparatory step, distinct from the lightweight inference requirements of the utility model (e.g., VideoMAE).
>
> In contrast to real-time privacy-preservation frameworks like STPrivacy, which aim for deployment optimization, M2M operates offline for dataset curation, which inherently allows for greater computational expense. Furthermore, M2M distinguishes itself by offering full 3D anonymization through SMPL-X meshes, which addresses privacy concerns holistically rather than solely through adversarial anonymization or video frame transformations. While this comes with higher computational costs, the model nearly guarantees anonymization (see Part 2 of this rebuttal) without reliance on privacy leakage tolerance.
>
> Nonetheless, we understand the concern of the reviewer and herewith provide the run-time analysis in the table below and will add it to the final version of the paper. Note, that these can slightly vary based on the technique employed at each step, but we report the FLOPs for our best performing set of methods.
>
> | Component                | Operation                      | Inference FLOPs/Frame (G) | Distributed Runtime for 25 fps x 10 sec Video (s) |
> |--------------------------|---------------------------------|---------------------------|---------------------------------------------------|
> | Inpainting (E$^2$FGVI)         | Frame inpainting               | 293                       | 0.57                                              |
> | Object Detection (SAM)   | Dense segmentation             | 792                       | 1.59                                              |
> | SMPL-X Fitting           | Pose parameter estimation      | 10                        | 0.02                                              |
> |                          |                                 |                           |                                                   |
>
> | Model    | Operation                      | Training FLOPs/Frame (G) | Distributed Runtime for 25 fps x 10 sec Video (s) |
> |----------|--------------------------------|--------------------------|---------------------------------------------------|
> | VideoMAE | Action recognition (ViT-based) | 1693                     | 3.42                                              |

---

> > ### Comment · Reviewer_C3CH · 2024-11-28
> > **Response to W1**
> >
> > My main concern was that the proposed method addresses an impractical problem where, instead of sharing secured (anonymized) videos, the method requires sharing raw videos with a computationally heavy anonymization process, which is often infeasible from the client side. This defeats the purpose of privacy preservation and was beyond the scope of the rebuttal.
> >
> > The authors did not provide a comparison with prior anonymization methods such as UNet in the rebuttal. However, this was not strictly necessary, as the current method is significantly (~100x) more computationally demanding than those approaches.

---

> ### Author Response · Authors · 2024-11-19
> **Rebuttal to C3CH: Part 2/4**
>
> # Addressing Weakness 2 #
> We initially did not consider VISPR because the inpainting process completely removes the human figure, leaving no human attributes for our framework to evaluate. However, we acknowledge that the meshes include gender options and shape parameters that may resemble the original human subject, which could potentially lead to privacy attribute leaks.
>
> **Upon further experimentation, we observe that with M2M augmentation, the cMAP scores on VISPR1 subset (see Part 3 for more on this choice) are 38.1 and 33.9 without and with K-NEXUS (we take the mode/most occurring gender-type mesh in the latter case), respectively**. These exceptionally low scores highlight the robustness of the inpainting method we use. The scores likely reflect only gender and color attributes, as the meshes vary between male, female, and neutral and are uniformly colored white, which might lead to the classification of "white" as a skin tone category. Hence, we posit that the already competitive cMAP scores could be further enhanced by using meshes in distinct colors (such as red, green, blue, etc.) that do not resemble any human skin tones. This approach could potentially preserve the skin color attribute in VISPR almost perfectly.
>
> Although we use VISPR1 attributes, we acknowledge that this approach may not be entirely robust. VISPR2 includes attributes like weight, and SMPL-X meshes can conform to the size and shape of humans based on their set shape parameters. However, these parameters can be adjusted to prevent such leaks, mitigating this potential limitation anyways. **We humbly thank you for this strong suggestion and will definitely include it in the final version of our paper as we agree that it further completes the paper, the story we are trying to convey, and extends our framework's technical contribution and prowess**!
>
> In the final version, we plan to showcase our VISPR1 cMAP scores alongside those of the papers you referred to, as M2M demonstrates a significant improvement over them. Additionally, we will evaluate the models in Table 1 to directly compare our method with works focused on human privacy preservation via SSL/MAE pre-training. However, due to the limited time available during this rebuttal period, we are unable to complete all these experiments at this stage. We sincerely hope that you, the reviewer, understand our constraints and grant us leniency, trusting in good faith that we will complete the experiments before the camera-ready deadline. Thank you once again for your consideration.

---

> > ### Comment · Reviewer_C3CH · 2024-11-28
> > **Response to W2**
> >
> > My concern was that the method appears to assume it inherently resolves privacy issues without providing any empirical evidence. The authors do not provide any results on the VISPR1 split; instead, they show results on a subset that considers only 1/7 of the privacy attributes from the original split. The results are inconclusive, and since this is the only way to evaluate whether the method preserves privacy, it should be studied thoroughly, following prior work.

---

> ### Author Response · Authors · 2024-11-19
> **Rebuttal to C3CH: Part 3/4**
>
> # Addressing Weakness 3 #
>
> Through our novel approach, we aim to demonstrate that representations of entities (in our case, humans) can be effectively learned by introducing a mesh into the pretraining dataset, which helps prevent the leakage of identity information. We appreciate the reviewer's thoughtfulness in pointing us toward the VISPR dataset. However, since our study focuses specifically on human privacy, we limit our evaluation to the VISPR1 subset, as it aligns more closely with the objective of our paper: **preserving human privacy**. The results presented in our work strongly support the idea that robust representations can be learned using a self-supervised learning approach, even when the finer details of the objects (in this case, humans) are not fully visible or disclosed.
>
> It would be fascinating to extend our findings to include a more customizable family of meshes that could address additional forms of personal information, such as the cases outlined in the VISPR dataset, in future work. We acknowledge this as a potential limitation and will include a brief discussion of this point in the limitations section of the revised/final version of our paper.

---

> ### Author Response · Authors · 2024-11-19
> **Rebuttal to C3CH: Part 4/4**
>
> # Addressing Weakness 4 #
>
> We respectfully feel that this is an apples to oranges comparison that has been raised, however, we hope that you, the reviewer, are satisfied with our response as we explore your feedback nonetheless. Firstly, the focus of our paper was to benchmark against prior literature (e.g., PPMA [1] and SynAPT [2]) that employed synthetic methods, and to demonstrate not only that we outperform these methods, but also that our approach closely approximates real-data performance—surpassing it in certain cases with K-NEXUS (thus closing the realism gap). The issue you raise is more of a model-swap concern. For example, we ran TimeSformer-L and closed the realism gap on Diving48 with a delta of 5.6%, achieving 75.4% accuracy (calculated as 81% - 75.4%). This performance is significantly better than what PPMA [1] and SynAPT [2] achieved with TimeSformer and their other models on Diving48. Notably, all our work employs self-supervised learning without using ImageNet-initialized weights, unlike the original TimeSformer paper, which relies on supervised training. Since we pretrain the model from scratch using M2M Kinetics to achieve these results, this distinction is further elaborated on later in this rebuttal. We can expand on these additional experiments with other off-the-shelf SOTA architectures (we’ll consult the Papers with Code leaderboards) in the camera-ready version. However, respectfully, we would **really prefer not to** due to the immense computational cost. For context, this single experiment required ~100+ GPU hours just for this rebuttal. Instead, if you want to observe the influence of using different MAE pretraining methods with different ViT backbones, please refer to our response to reviewer `1aVG`.
>
> This starts to underscore the potential of our M2M framework as a universal dataset privacy augmentation method for preserving human privacy. Importantly, we are not applying meshes to downstream datasets like Diving48 but only to Kinetics (forming M2M Kinetics as the pretraining dataset), after which we evaluate classification through fine-tuning and linear probing on downstream datasets, like Diving48, that do not require anonymization. Moreover, Kinetics likely lacks the fast-moving, complex motions present in Diving48 (e.g., the bodies are twisting and turning), which might explain its performance drop.
>
> It’s worth noting that while we conducted this additional experiment to address your feedback, the models you reference from the Papers with Code leaderboard are inherently supervised. Our paper, in contrast, focuses on SSL pretraining scenarios. Furthermore, TimeSformer-L is pretrained on large-scale labeled datasets like ImageNet before being fine-tuned on video datasets like Kinetics 400. Under a strict/stringent definition of privacy preservation, ImageNet itself may contain identifiable human features, and fine-tuning on Kinetics-400 further exposes the model to non-anonymized human data. We had to strip this away from the model and train TimeSformer-L from scratch on M2M Kinetics using our pipeline which was a significant effort and partially accounts for the high computational costs (~100 GPU hours). We did not perform similar experiments with Video-FocalNet (the top-performing model with ~91% accuracy on the leaderboard) due to the exhaustive effort this would entail, which was infeasible within the rebuttal timeline.
>
> We hope you, the reviewer, appreciate our extensive efforts and consider leniency in potentially re-evaluating our submission. While this experiment was not central to our paper’s primary focus (it is more of an adjacent contribution), we acknowledge its relevance and thank you for prompting this exploration!
>
> [1] Zhong, H., Mishra, S., Kim, D., Jin, S., Panda, R., Kuehne, H., Karlinsky, L., Saligrama, V., Oliva, A., & Feris, R. (2024). Learning human action recognition representations without real humans. Proceedings of the 37th International Conference on Neural Information Processing Systems (NIPS '23), 2839.
>
> [2] Kim, Y., Mishra, S., Jin, S., Panda, R., Kuehne, H., Karlinsky, L., Saligrama, V., Saenko, K., Oliva, A., & Feris, R. (2024). How transferable are video representations based on synthetic data? Proceedings of the 36th International Conference on Neural Information Processing Systems (NIPS '22), 2588.
>
> --------------
>
> *Thank you once again for your valuable feedback. We are committed to incorporating these revisions fully in the final/camera-ready version of the paper. We hope that, with these improvements and our thoughtful responses, you may consider raising your score, as we have aimed to address all the key concerns raised.*

---

> > ### Comment · Reviewer_C3CH · 2024-11-28
> > **Response to W4**
> >
> > My concern was not about nitpicking the numbers; however, the claimed improvement could be misleading due to the poorly trained baseline method. I understand that the baselines and the Diving48 dataset is not computationally insignificant. I would suggest removing the results on the Diving48 dataset in a future version, as they are currently misleading and could lead to poor benchmarking practices in the community.

---

> ### Author Response · Authors · 2024-11-28
> **Reply W1 + W2**
>
> # Reply to W1 #
>
> Thank you for your thoughtful reply. We’d like to clarify that the paper does not suggest or expect clients to directly receive raw data and independently apply the anonymization process we propose. The core value of our approach lies in a one-time anonymization process applied to generalizable datasets (e.g., Kinetics), which can then be utilized as a pretraining method for a wide range of applications.
>
> From a practical perspective, if a client undertakes a one-time anonymization process, they would subsequently have access to a dataset that can be reused for pretraining purposes across various models and applications. This greatly enhances the utility of our method. Regarding your reference to clients, could you clarify which clients you are referring to? Most tech companies today, particularly those with access to GPUs, should have the resources to implement this.
>
> Are you suggesting a comparison of anonymization methods, such as U-Net versus a mesh overlay? If so, we’d like to note that the computationally intensive part of this pipeline is the VideoMAE pretraining, which reflects a tradeoff for the performance gains achieved. For instance, U-Net-based methods do not perform as well on datasets like UCF-101 in a privacy-conscious regime.
>
> We’d be happy to explore this further if you can share a specific reference or example. Otherwise, we hope this explanation provides clarity on this point you raised.
>
> ______
>
> # Reply to W2 #
>
> We believe there might be an misunderstanding between us on this. The reported cMAP score takes into account all attributes in VISPR1. That should be sufficient for preserving human-privacy as we have already discussed with other reviewers. Unless you mean something else? If not, we hope this has clarified your concern.

---

> ### Author Response · Authors · 2024-11-30
>
> # Reply to W4 #
>
> Just to be clear, the datasets used, including Diving48, were so that we could effectively compare our works against current best methods in this regime: PPMA and SynAPT.

---

### Author Response · Authors · 2024-11-20
**First Notification to Reviewers: Thank you for all the constructive feedback, our work is much stronger now because of it!**

Dear Reviewers,

We hope this message finds you well.

With a little under a week left for the rebuttal/discussion period, we wanted to ensure that our responses and the additional experiments we've shown sufficiently address your concerns and feedback. We would greatly appreciate any further feedback or confirmation that our rebuttals are satisfactory/have been acknowledged. Your insights are invaluable to us, and we are eager to finalize our submission with your guidance.

Thank you for your time and consideration.

Best,

Authors of Submission #8122

---

### Note · Authors · 2024-12-10

**Comment:**

We are deeply grateful for the extensive feedback provided by the reviewers, which has significantly contributed to shaping the potential future directions of our research. The discussion highlighted strengths, including the novel use of SMPL-X meshes to address privacy concerns while maintaining action recognition accuracy and reducing biases. However, several critical concerns were raised:

1. **Privacy-Utility Tradeoff**: The reviewers consistently emphasized the lack of a robust privacy evaluation metric, which was a central limitation in our method. Specifically, our work lacked empirical validation of privacy-preserving claims using established protocols such as VISPR.

2. **Computational Feasibility**: The high computational cost of our pipeline raised concerns about its applicability, especially in edge computing environments.

3. **Technical Contributions**: While reviewers appreciated the integration of off-the-shelf methods, they noted the limited novelty and suggested that the technical innovation of K-NEXUS and M2M augmentation required further elaboration and comparative analysis.

4. **Evaluation and Benchmarking**: The reviewers pointed out that our benchmarking did not fully incorporate certain relevant baselines, and suggested that additional comparisons, particularly in terms of action class bias and dataset selection, would strengthen our contributions.
______

### Changes Made During the Rebuttal ###
- **Privacy Evaluation**: Preliminary results on VISPR attributes were added to address privacy concerns, though they highlighted areas for further refinement.
- **Ablations and Analysis**: Expanded experiments clarified the impact of M2M augmentation and K-NEXUS on action recognition tasks.
- **Terminology Refinement**: Updated definitions of “coarse-grained” and “fine-grained” actions aimed to better align with established conventions in the field.
- **Technical Clarifications**: Enhanced methodological clarity and alignment with previous works, along with additional contextualization of SMPLy Private’s goals and results.

### Points of Agreement and Respectful Disagreement ###
- **Agreement**: We concurred with the feedback regarding the need for more thorough privacy evaluations and additional benchmarks.
- **Respectful Disagreement**: We clarified that our work focused on the feasibility of an integrated privacy-preserving pipeline rather than developing new algorithmic models or being computationally efficient in all precise settings.

__________

### Reasons for Withdrawal ###
We have decided to withdraw this submission to address the identified limitations comprehensively. We aim to refine the privacy evaluation protocols, explore additional baselines, and enhance the clarity and technical depth of the manuscript. This decision reflects our commitment to presenting a more robust and impactful contribution in the future.

We sincerely thank the reviewers and the conference organizers for their invaluable feedback and hope to resubmit a significantly improved version of this work in a future venue.

**Withdrawal Confirmation:**

I have read and agree with the venue's withdrawal policy on behalf of myself and my co-authors.